

# Ozone deposition measurements over wheat fields in the North China Plain: variability and related factors of deposition flux and velocity

Xiaoyi Zhang[1,2], Wanyun Xu[1, *], Weili Lin[3], Gen Zhang[1], Jinjian Geng[4,5], Li Zhou[4,5], Huarong Zhao[4,5], Sanxue Ren[4,5], Guangsheng Zhou[4,5], Jianmin Chen[2] and Xiaobin Xu[1, *]

[1] State Key Laboratory of Severe Weather, Key Laboratory for Atmospheric Chemistry, Institute of Atmospheric Composition, Chinese Academy of Meteorological Sciences, Beijing, China, 100081
[2] Department of Atmospheric and Oceanic Sciences, Fudan University, Shanghai, China, 200433
[3] College of Life and Environmental Sciences, Minzu University of China, Beijing, China, 100081
[4] State Key Laboratory of Severe Weather, Institute of Agricultural Meteorology, Chinese Academy of Meteorological Sciences, Beijing, China, 100081
[5] Hebei Gucheng Agricultural Meteorology National Observation and Research Station, Baoding, China, 072656

*Correspondence to*: Wanyun Xu (xuwy@cma.gov.cn) and Xiaobin Xu (xiaobin_xu@189.cn)

**Abstract.** Ozone ($O_3$) deposition is closely related to air quality, ecosystem and climate changes. Due to the instrument and method shortage, $O_3$ deposition was less observed and investigated in China, experiencing significantly increasing $O_3$ exposure. Here, we conducted a comprehensive measurement of $O_3$ deposition over the wheat canopy at a typical polluted agricultural site in the North China Plain using a newly developed relaxed eddy accumulation system. For the main wheat growing season in 2023, $O_3$ deposition flux and velocity ($V_d$) averaged -0.25 ± 0.39 µg m$^{-2}$ s$^{-1}$ and 0.29 ± 0.33 cm s$^{-1}$, respectively. Daytime $V_d$ (0.40 ± 0.38 cm s$^{-1}$) was obviously higher than in the nighttime (0.17 ± 0.26 cm s$^{-1}$). The temporal changes of $V_d$ were mainly determined by crop growth, with predominant contribution of stomatal uptake. Both daytime and nighttime $V_d$ exhibited significant increases with decreasing relative humidity, and increasing friction velocity and soil water content, enhanced by higher leaf area index. With rapid increases of soil moisture, simultaneous and following overall increments in $V_d$ were detected, attributed to remarkably strengthening $O_3$ stomatal uptake under increased stomatal conductance and extended opening to the night, and more non-stomatal $O_3$ removal at night resulted from strengthened soil NO emission at moist conditions. This study confirms the leading effects of crop growth on $O_3$ deposition modulated by environmental conditions and the non-negligible influences of nocturnal plant activities, and emphasizes the needs for $O_3$ deposition observation over different surfaces and accurate evaluation of $O_3$ agricultural impacts based on deposition fluxes.

## 1 Introduction

Surface ozone ($O_3$) is a key secondary air pollutant, generated in photochemical reactions involving volatile organic compounds (VOCs), nitrogen oxides ($NO_x$ = NO ＋$NO_2$), etc. (Seinfeld et al., 2006). Over the past two decades, China's rapid economic development and increasing anthropogenic emissions of $NO_x$ and VOCs have led to significantly upward



trends in $O_3$ concentrations (Monks et al., 2015; Li et al., 2019; Lu et al., 2020; Xu et al., 2020), especially in the North China Plain (NCP) (Tai et al., 2014; Ma et al., 2016; Wang et al., 2017; Lu et al., 2020; Xu, 2021; Wang et al., 2022; Lyu et al., 2023). Dry deposition plays one of the key roles in removing surface $O_3$ (e.g., Tang et al., 2017) and in the budget of

tropospheric $O_3$ (Lelieveld and Dentener, 2000). Over vegetated areas, stomatal and non-stomatal uptake of $O_3$ may represent major part of the total dry deposition (Fowler et al., 2001). Uptake of higher amount of $O_3$ may cause a series of hazardous oxidative reactions, damaging vegetation and threatening crop quality and production (Ainsworth, 2017; Harmens et al., 2018; Mills et al., 2018; Feng et al., 2019b). In addition, $O_3$ deposition to the ground surfaces (including soil, snow and water) is closely related to tropospheric chemistry, air quality and ecosystems (Clifton et al., 2020a; Stella et al., 2019;

Helmig et al., 2012; Stocker et al., 1995). It is estimated that $O_3$ dry deposition contributes about 20% to the annual global tropospheric $O_3$ loss (Lelieveld and Dentener, 2000; Wild, 2007; Hardacre et al., 2015). Under the rapid expansion of population and growing demands for food, China has become the largest crop producer, as well as importer (Dong et al., 2021). $O_3$ deposition is thus of great importance and its accurate quantification is urgently needed to evaluate the impact of increasing $O_3$ levels on agricultural production, ecological environment, air quality, human health, and global climate.

$O_3$ deposition has been measured over various ecosystems, including forest, grassland, cropland and bare soil environments (Table S1), in order to understand deposition mechanisms and evaluate its potential effects (Stella et al., 2019; Xu et al., 2018; Zhu et al., 2015; Helmig et al., 2012; Mészáros et al., 2009; Lamaud et al., 2009; Coyle et al., 2009). However, the deposition processes are controlled by various biotic (stomatal uptake) and abiotic (non-stomatal removal) activities that are simultaneously modulated by the environmental factors. The relative contributions of stomatal and non-

stomatal $O_3$ deposition varied with land cover, plant species and growth stages, as well as environmental factors. Stomatal uptake of $O_3$ depends on the opening and closure of stomata on leaf surfaces. For example, the fraction of diurnal maximum stomatal $O_3$ deposition over boreal forests ranged from 56 to 74% (Rannik et al., 2012), while only accounting for 31.2% in a wheat field (Xu et al., 2018). Non-stomatal resistance of $O_3$ decreased with the increasing temperature and friction velocity, and was ~ 50% lower under wet conditions than under dry conditions over the same potato canopy (Coyle et al., 2009). Thus,

$O_3$ deposition is dominated by distinct deposition processes over different surfaces in different environments.

Currently, the eddy covariance (EC) method is the most commonly used micrometeorological technique for measuring vertical fluxes and is also regarded as the most ideal method (Businger, 1986; Baldocchi et al., 1988). However, it requires robust fast-response measurement instruments ($\geq$ 10 Hz), which limits the flux measurements of various reactive gases due to the lack of high frequency detection techniques (Hicks and Wesely, 1978; Muller et al., 2009). The flux-gradient (FG)

approach is the most important alternative method for $O_3$ flux measurements (Stella et al., 2012; Loubet et al., 2013; Wu et al., 2015), which is based on the flux-gradient relationship (K-theory) (Baldocchi, 1988). The assumption of flux-gradient relationship is dependent on surface roughness and the photochemical reactions of $O_3$ and its precursors, which does not hold over rough heterogeneous surfaces (such as forests) and when $O_3$ formation reaction rates exceed turbulent mixing rates (Raupach and Thom, 1981; Vilà-Guerau De Arellano and Duynkerke, 1992). Besides, FG needs simultaneous $O_3$

concentration measurements at several heights (2 ~ 4) and accordingly requires several parallel $O_3$ analyzers (Loubet et al.,



2013). These relatively high requirements of instruments have more or less limited the application of traditional micrometeorological methods to the measurements of $O_3$ flux.

The relaxed eddy accumulation (REA) method is an alternative method for observing the air-surface exchange of interested substances over ecosystems, which is based on the same physical principle as EC (Desjardins, 1977; Businger and Oncley, 1990; Pattey et al., 1993). The fluxes in REA systems are obtained by accumulating and accurately measuring the air associated with updrafts and downdrafts at a constant flow rate in two separate reservoirs (Businger and Oncley, 1990). REA methods have been wildly applied in flux measurements of various species, such as biogenic VOCs (Mochizuki et al., 2014; Moravek et al., 2014), peroxyacetyl nitrate (Moravek et al., 2014), reduced sulfur gases (Xu et al., 2002), HONO (Ren et al., 2011), aerosols (Matsuda et al., 2015; Xu et al., 2021), $NH_3$ and Hg (Osterwalder et al., 2016). To the best of our knowledge, the REA method has not been applied in $O_3$ deposition flux measurements so far.

Although many regions in China have been experiencing severe $O_3$ pollution during growing seasons, measurements of $O_3$ flux over crop fields in the country have only sporadically reported, which were made using either chamber techniques (e.g., Tong et al., 2015) or micrometeorological approaches (Zhu et al., 2014; Zhu et al., 2015; Xu et al., 2018). In this study, we developed a new REA flux system and applied it to obtain $O_3$ deposition fluxes over wheat fields in the NCP during the springtime growing season. Based on these in-situ observations, we evaluated the feasibility of $O_3$ flux measurements using the REA method, analyzed the variation characteristics of $O_3$ deposition during the wheat growing season, identified the key drivers in the variability of daytime and nighttime $O_3$ deposition during distinct crop growth stages and different environmental conditions, respectively.

## 2 Observation and method

### 2.1 Site description

The flux observations were conducted at the Gucheng site (39°08'N, 115°40'E, GC), an integrated ecological-meteorological observation and research station of the Chinese Academy of Meteorological Science, located 35 km to the northeast of urban Baoding City, Hebei Province, and 100 km southwest to urban Beijing. The site is surrounded mainly by flat, irrigated high-yield agricultural lands in the northern part of the NCP (see Fig. 1a in Zhang et al., 2022b). The fields within and surrounding the yard of GC are on a winter wheat-summer maize rotation, which is typical in Northern China. Observations at the site have revealed good regional representativeness of the polluted agricultural areas in the NCP (Lin et al., 2009; Xu et al., 2019; Kuang et al., 2020; Zhang et al., 2022a).



## 2.2 Relaxed eddy accumulation (REA) technique

### 2.2.1 Theory

A homemade relaxed eddy accumulation system for $O_3$ dry deposition measurements (REA-$O_3$ flux system) was deployed at GC. In the REA-$O_3$ flux system, conditional sampling is conducted according to the direction of vertical wind ($w$), which separates sampled air into updraft and downdraft reservoirs at a constant flow rate. The vertical fluxes of $O_3$ ($F_{O_3}$, in µg m$^{-2}$ s$^{-1}$) are calculated by the concentration differences between two reservoirs following Eq. (1):

$$F_{O_3} = \overline{w'\,c'} = b\sigma_w\left(\overline{c^+} - \overline{c^=}\right),\qquad\qquad\text{Eq. (1)}$$

where $\sigma_w$ is the standard deviation of vertical wind (in m s$^{-1}$); $\overline{c^+}$ and $\overline{c^=}$ are the averaged $O_3$ concentrations in the updraft and downdraft reservoirs, respectively (in µg m$^{-3}$); $b$ is the eddy accumulation coefficient and is obtained from $CO_2$ flux measured using the EC method and calculated using Eq. (2):

$$b = \frac{\overline{w'\,CO_2'}}{\sigma_w\left(\overline{CO_2^+} - \overline{CO_2^=}\right)},\qquad\qquad\text{Eq. (2)}$$

where $\overline{CO_2^+}$ and $\overline{CO_2^=}$ are averaged $CO_2$ concentration observed under upward and downward vertical winds, respectively
(in mg m$^{-3}$), $\overline{w'\,CO_2'}$ represents the EC $CO_2$ flux (in mg m$^{-2}$ s$^{-1}$). Based on the measurements from February to June, $b$ revealed an average ± standard deviation of 0.55 ± 0.09, ranging from 0.16 to 0.80.

$O_3$ deposition velocity ($V_d$, cm s$^{-1}$) is estimated based on $O_3$ flux and concentrations using Eq. (3):

$$V_d = -\frac{F_{O_3}}{C_{O_3}} \times 100,\qquad\qquad\text{Eq. (3)}$$

where $C_{O_3}$ is the 30 min averaged $O_3$ concentration (in µg m$^{-3}$).

### 2.2.2 System setup and verification

The setup of the REA-$O_3$ flux system is depicted in Figure 1. A 3-D sonic anemometer (CSAT3, Campbell Scientific Inc., USA) was used for measuring the three wind components ($u$, $v$, $w$) at 10 Hz, which was amounted at the height of 4.5 m on an eddy covariance tower. The inlet (1/8" OD Teflon tubing) of the REA system was installed in the center of the anemometer. The 10 Hz wind signals, together with the signals from the $CO_2$/$H_2O$ analyzer (Li-7500, LI-COR, Inc., USA),
were collected by a datalogger (CR1000, Campbell Scientific Inc., USA) and sent to a PC. The wind signals were processed by a program written in Python, which also sent switch command to two fast-response 3-way solenoid valves based on the vertical wind direction. Sample air was thus pumped alternatively into sample tubes (1/4" OD Teflon) for updraft or downdraft based on the direction of instantaneous vertical winds and analyzed by two UV photometric $O_3$ analyzers (TE 49i, Thermo Fisher Scientific Inc., USA) installed at the ends of updraft and downdraft channels, respectively. To ensure the
stability of airflow in the $O_3$ analyzers and sampling system, zero air was supplied to the channel that was not sampling ambient air. In addition, both sampling tubes were bypassed to increase the inlet sample flow and avoid the axial mixing before the solenoid valves. The linear velocity of air sample in the inlet tubing was set to 22 m s$^{-1}$, and airflow was at the



turbulent state with a Reynolds number over 2300. The estimated residence time from system inlet to the valves was 18 ms

and the REA system was working at a sample frequency of 10 Hz. The $O_3$ analyzers recorded 1-minute averaged $O_3$

concentrations, which were downloaded by the PC. The actual averaged $O_3$ concentrations under updraft and downdraft

conditions were calculated according to Eq. (4) using the sample time, sample flow and 1-minute averaged $O_3$ concentrations:

$$\bar{c} = \frac{\sum_{i=1}^{i=30} c_i \times flow_i}{\sum_{i=1}^{l=30} flow_i \times t_{sample\ gas,i}},$$                                                        Eq. (4)

where $c_i$ is the 1-minute averaged $O_3$ concentration (in μg m$^{-3}$), $flow_i$ is the 1-minute averaged sample flow (in SLPM)

measured by the mass flow meter (MFM), and $t_{sample\ gas,i}$ the real time for analyzing air sample within $i$th minute (in

fraction of a minute).

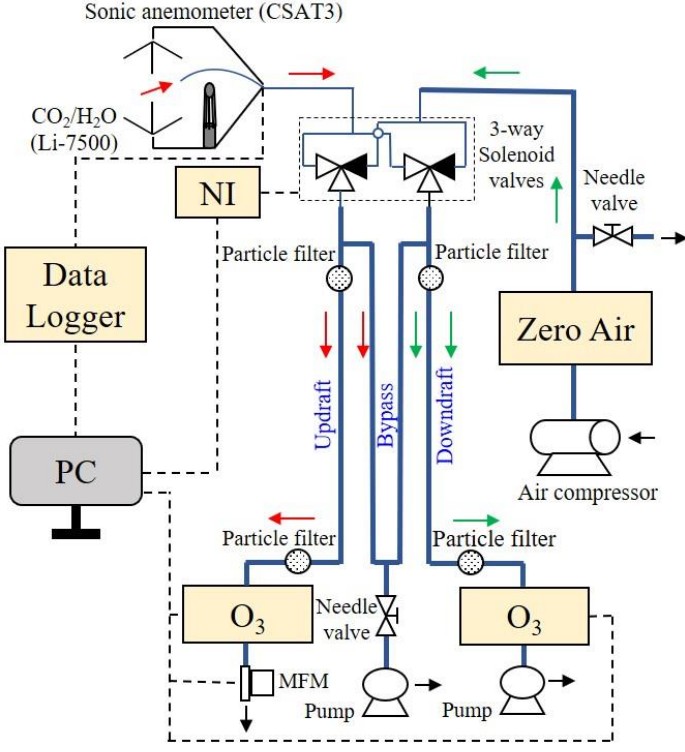

**Figure 1: Schematic of the REA-$O_3$ flux system ($w > w_0$).**

To increase the concentration difference between updraft and downdraft, a wind speed threshold called the wind dead-

band ($w_0$) was used in the REA system to discard air samples associated with $w$ close to 0. The application of $w_0$ promotes

the sampling of larger eddies that contribute more to gas fluxes. If a proper $w_0$ is used, the eddy frequency spectrum shifts to

the low frequencies in the sample, but does not cut off all the high-frequency signals, only filtering out samples with small

vertical displacements that have relatively small impacts on the overall flux (Bowling et al., 1999; Held et al., 2008; Tsai et

al., 2012; Moravek et al., 2014; Grelle and Keck, 2021). Conditional sampling using $w_0$ also prolongs the lifetime of the fast-





response solenoid valves (Pattey et al., 1993) and effectively avoids sampling error around $w = 0$ due to the limitation of the

sonic anemometer (Grelle and Keck, 2021). In the REA-$O_3$ system, we adopted fixed wind dead-bands during the daytime (from 08:00 to 18:00 Local Time (LT), $w_0 = 0.05$ m s$^{-1}$) and night (from 19:00 to 07:00 LT, $w_0 = 0.01$ m s$^{-1}$), respectively, considering that wind speeds during the daytime were generally larger than those at night. The concentration difference increased with $w_0$ and led to an increase in measured fluxes. Figure S1 shows the comparison of REA $CO_2$, $H_2O$ and heat fluxes under conditional sampling adopting $w_0$ and full sampling, respectively, calculated using raw EC data with a constant

$b$ of 0.60 (Businger and Oncley, 1990). The two flux datasets revealed excellent correlations, with a correlation coefficient close to 1, proving the reliability and stability of the REA flux measurement system. Compared with the fluxes without $w_0$, $CO_2$, $H_2O$ and heat fluxes with $w_0$ exhibited similar small overestimations, reaching around 10-13% during the daytime and 4-10% at night, which were comparable with the influence of a dynamic dead-band ($w_0 = \frac{\sigma_w}{0.35}$) in Grelle and Keck's (2021) REA system for $H_2O$, $CO_2$, $CH_4$, $N_2O$ flux measurements. This indicates that adopting the wind dead-band in our REA

system only had marginal impact on observed fluxes.

      To identify potential flux errors induced by any differences between updraft and downdraft channels in the REA-$O_3$ system (including valves, sample tubes and $O_3$ analyzers), we checked the parallel of sampling system by simultaneous measurements of ambient $O_3$ from the inlet. As shown in Figure S2, $O_3$ data points obtained from the two channels all aligned close to the 1:1 line (slope = 1.02), suggesting that the difference in measured $O_3$ was minimal between updraft and

downdraft and its impact on flux measurement can be ignored. Moreover, synchronous multipoint calibrations of the two channels were conducted monthly. Different $O_3$ concentrations generated by an $O_3$ calibrator (TE 49C PS, Thermo Fisher Scientific Inc., USA) were introduced into the system from the zero air inlet and simultaneously measured by the two $O_3$ analyzers.

      To further verify the reliability of the REA system, we also compared the fluxes data derived from the REA technique

with $w_0$ and EC theory. For $CO_2$, $H_2O$ and heat, the averaged ratios of REA to EC fluxes were all slightly larger than 1, indicating small overestimates in the REA flux measurement system, which were expected due to the use of $w_0$. Most of the flux data maintained high consistency with correlation coefficients close to 1 (Figure S3), confirming that the REA system performed reliably under most conditions.

### 2.3 Field measurements and ancillary data

Measurements of $O_3$ flux were conducted during the main wheat growing season in 2023, from the late of dormancy stage (13 February 2023) to wheat harvest (18 June 2023). Ancillary data were obtained for further analysis. Meteorological variables including air temperature ($T_{Air}$), relative humidity (RH), precipitation, soil temperature ($T_{Soil}$) and volumetric water content (soil VWC) at 20 cm, global solar radiation (G), photosynthetic active radiation (PAR) and sun elevation angle were measured by an automatic weather station at GC. The 30 min $CO_2$, $H_2O$, heat and momentum fluxes were measured by the

EC system, which includes a 3-D sonic anemometer, an open path $CO_2/H_2O$ analyzer and a datalogger. The friction velocity



($u_*$) was calculated using the three wind components ($u$, $v$, $w$) following Eq. (5), and the vapor pressure deficit (VPD) was estimated as in Eq. (6).

$$u_* = \left( \overline{u'w'}^2 + \overline{v'w'}^2 \right)^{1/4} , \qquad \text{Eq. (5)}$$

$$VPD = (1 - \frac{RH}{100}) \times 611.2 \times exp(\frac{17.62 \times T_{Air}}{243.12+T_{Air}}) \div 100 , \qquad \text{Eq. (6)}$$

The Leaf Area Index (LAI) and the Fraction of Photosynthetically Active Radiation (FPAR) were obtained from the Moderate Resolution Imaging Spectroradiometer (MODIS) Level 4 product (MCD15A3H) with a spatial resolution of 500 m and temporal resolution of 4 days (Myneni et al., 2015).

NO$_x$ (NO/NO$_2$/NO$_x$) concentrations were monitored from 18 March to 2 June by a NO-NO$_2$-NO$_x$ Trace Level Analyzer (Model 42C-TL, Thermo Fisher Scientific Inc., USA). Multipoint calibrations of NO$_x$ were made using a NO standard gas
obtained from National Institute of Metrology, Beijing, China.

### 2.4 Stepwise multiple linear regression (MLR) model

Stepwise multiple linear regression (MLR) models were applied to identify the key environmental factors influencing O$_3$ deposition in the daytime (sun elevation angle > 0°) and nighttime (sun elevation angle < 0°), respectively. MLR is a commonly used approach to describe the relationship between air pollution and its influencing factors (Zhang et al., 2022b;
Han et al., 2020; Fu and Tai, 2015; Rannik et al., 2012). The stepwise MLR model takes the following form:

$$y = \beta_0 + \sum_{k=1}^{n} \beta_k x_k , \qquad \text{Eq. (7)}$$

where $y$ is the observed V$_d$, $x_k$ is the selected normalized environmental parameter, $\beta_0$ is the regression constant, $\beta_k$ is the regression coefficient and $n$ is the number of selected term. $\beta_k$ is determined by a forward stepwise method to add and delete terms to obtain the best model fit based on Akaike Information Criterion (AIC) statistics (Venables and Ripley, 2003).
Environmental parameters including seven meteorological and soil factors (T$_{Air}$, RH, VPD, $u_*$, T$_{Soil}$, soil VWC and PAR) and two crop related factors (LAI, FPAR) were considered during daytime, while PAR was unaccounted for during nighttime. The selected variables in the stepwise MLR were considered to be the environmental factors critical for O$_3$ deposition at GC during the wheat growing season.

## 3 Results and discussion

### 3.1 Meteorological conditions

Figure 2 shows the temporal variations of daily meteorological and soil conditions during the whole period. Air and soil temperatures gradually increased from the lowest values (T$_{Air}$: -0.8 °C, T$_{Soil}$: 2.9 °C) in February to highest ones (T$_{Air}$: 30.2 °C, T$_{Soil}$: 30.9 °C) in June. RH varied around a higher level before the latter part of April and a lower level after that, with an average of 64 ± 17 %. Calculated $u_*$ fluctuated in the range of 0.05-0.3 m s$^{-1}$, with an average of 0.17 ± 0.04 m s$^{-1}$.



Daily VPD was relatively stable during February-early April, with an average of 5.9 ± 4.0 hPa., and rose obviously afterwards, reaching an averaged value of 19.9 ± 7.4 hPa during May to June. Soil VWC kept flat before the middle of April and showed dramatical boosts caused by strong precipitation or irrigation events during 9-10, 20-21, 28-29 April and 19-20 May, followed by slow declines due to evapotranspiration under higher temperatures. Both PAR and G exhibited great fluctuations with slight increases from February to June.

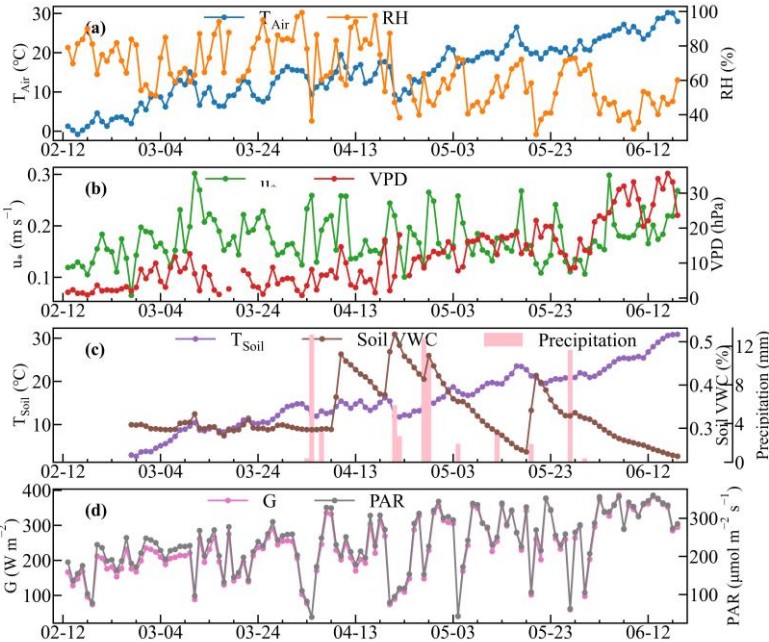


**Figure 2: Daily meteorological and soil conditions from 12 February to 18 June 2023: (a) T$_{Air}$ and RH, (b) u$_*$ and VPD, (c) precipitation, T$_{Soil}$ and soil VWC, (d) G and PAR.**

### 3.2 O$_3$ flux, deposition and concentration

  Totally 2728 pairs of O$_3$ deposition flux and velocity were obtained for the wheat growing season, which are presented in
Figure 3, along with 30-min and daily average O$_3$ concentrations. O$_3$ deposition flux averaged -0.25 ± 0.39 µg m$^{-2}$ s$^{-1}$, with a larger deposition during daytime (-0.39 ± 0.45 µg m$^{-2}$ s$^{-1}$) and a smaller one during nighttime (-0.08 ± 0.21 µg m$^{-2}$ s$^{-1}$). The largest deposition flux (-3.20 µg m$^{-2}$ s$^{-1}$) was measured at the noontime of 29 April, while the largest emission flux (0.14 µg m$^{-2}$ s$^{-1}$) at the midnight of 15 March. Daytime V$_d$ averaged 0.40 ± 0.38 cm s$^{-1}$ and was distinctly higher than nighttime ones (0.17 ± 0.26 cm s$^{-1}$). The averages of daytime and nighttime V$_d$ obtained in this study were comparable to those from
previous EC-based observations (0.42 cm s$^{-1}$ and 0.14 cm s$^{-1}$) during the wheat growing season and higher than those (0.29 cm s$^{-1}$ and 0.09 cm s$^{-1}$) during the maize growing season (Table S1) in Shangdong Province, China (Zhu et al., 2015; Zhu et al., 2014). V$_d$ averaged 0.29 ± 0.33 cm s$^{-1}$ over the whole observation period, ranging from -0.39 cm s$^{-1}$ to 2.65 cm s$^{-1}$. The average O$_3$ deposition velocities observed over the wheat canopy did not show substantial differences from those previously





reported for grasslands (Mészáros et al., 2009; Coyle, 2005), forests (Wu et al., 2015; Rannik et al., 2012) and bare soil
(Stella et al., 2019) (Table S1), considering the large uncertainties of reported mean values. $O_3$ concentrations over the wheat
canopy were significantly enhanced after April, with an overall average of $61.8 \pm 34.6$ µg m$^{-3}$. In general, $O_3$ deposition
velocities were more pronounced from mid-April to late-May (Figure 3b), when wheat was growing vigorously, while
deposition fluxes were higher after late May due to overall higher $O_3$ concentration (Figure 3a, c). Thus, $O_3$ concentration
was more determinative of $O_3$ deposition flux than $V_d$ on longer timescales.

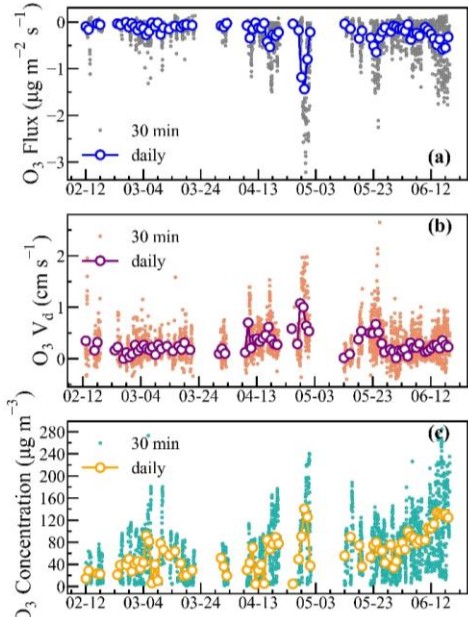


**Figure 3: Timeseries of $O_3$ deposition flux (a), $V_d$ (b) and concentration (c) during the wheat growing season.**

The averaged diurnal patterns of $O_3$ deposition flux, velocity and $O_3$ concentration are depicted in Figure 4. With plant
stomatal conductance and atmospheric turbulence increasing after sunrise, $O_3$ deposition rapidly rose during the morning
(06:00-10:00). Deposition flux and velocity both reached their peaks (-0.62 µg m$^{-2}$ s$^{-1}$ and 0.54 cm s$^{-1}$) by 13:00, when
stomatal conductance also reached a diurnal maximum (Rannik et al., 2012; Otu-Larbi et al., 2021). $O_3$ deposition quickly
decreased from 14:00 to 18:00 despite of high levels of $O_3$ (Figure 4). Nighttime $O_3$ deposition remained at relatively low
levels and exhibited weak changes, with an averaged flux and $V_d$ of $-0.09 \pm 0.04$ µg m$^{-2}$ s$^{-1}$ and $0.17 \pm 0.02$ cm s$^{-1}$,
respectively. Therefore, diel variations in $O_3$ deposition over the wheat fields were mainly driven by the stomatal opening,
with $O_3$ deposition velocity being decisive of deposition flux diurnal variations.





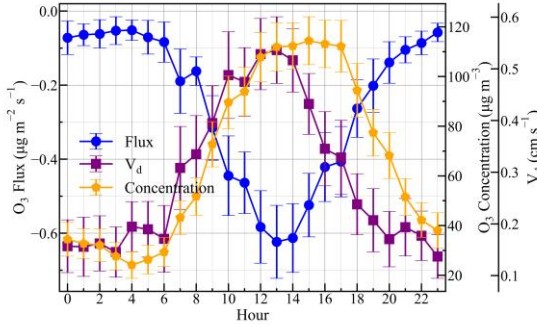


**Figure 4: Diurnal variations of O₃ deposition flux, Vd and concentration during the wheat growing season, with error bars representing average ± 0.2 × standard deviation values.**

### 3.3 O₃ deposition in different stages of wheat growth

To investigate the influences of wheat growth on O₃ deposition, the entire growth season was divided into three stages:

Over-Wintering (O-W, 13 February-5 March), Green-Flowering (G-F, 6 March-28 May) and Ripening-Harvest (R-H, 29 May-18 June), according to the winter wheat phenology at GC (Table S2). During the O-W stage, wheat was in dormancy, and leaves had not begun to turn green, with LAI < 0.5 (Figure 5b). $V_d$ in the O-W stage barely changed, exhibiting a low average of $0.20 \pm 0.28$ cm s$^{-1}$ and a median of $0.12$ cm s$^{-1}$ (Table 1). Wheat grew vigorously in the G-F stage, with LAI exhibiting rapid increases until the early flowering stage, after which LAI gradually decreased (Figure 5b). O₃ deposition

varied nearly in synchronization with LAI and wheat growth, with $V_d$ reaching a peak when LAI exceeded 4 during the G-F stage (Figure 5a), reaching highest daytime and nighttime averages of $0.46 \pm 0.41$ cm s$^{-1}$ and $0.24 \pm 0.28$ cm s$^{-1}$, respectively (Table 1). Afterwards, with the maturing of wheat and the aging of leaves in the R-H stage, $V_d$ quickly dropped back to a low average level of $0.20 \pm 0.25$ cm s$^{-1}$, similar to that observed in the O-W stage. It can be seen that the temporal variation of O₃ deposition velocity over wheat was predominantly determined by crop growth at GC. As for the deposition flux, both

the daily and daytime averaged fluxes during the G-F stage were comparable with those in the R-H stage (Table 1), which can be attributed to the high O₃ concentrations in the summer months (Zhang et al., 2022a; Lin et al., 2009). Although nighttime O₃ concentration during the G-F stage was also 58% lower than that in the R-H stage, nighttime O₃ deposition flux during the G-F stage was still the highest among the three stages, which was related to the remarkably high nighttime deposition velocities during the G-F stage (Table 1).





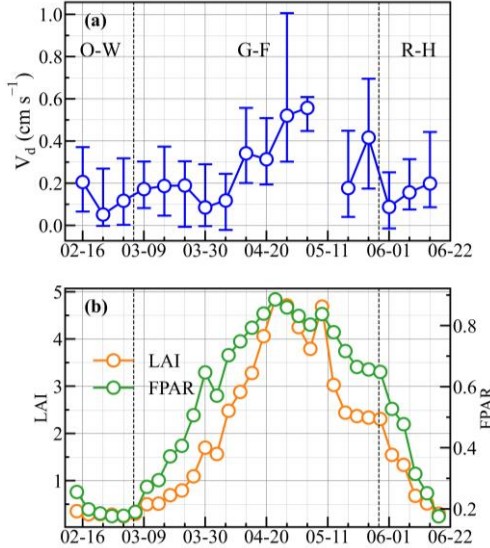

**Figure 5: (a) O₃ Vₐ, (b) LAI and FPAR in different wheat growing stages. The circles and error bars in (a) denote the weekly medians and quantiles of Vₐ, respectively. O-W, G-F and R-H represent Over-Wintering, Green-Flowering and Ripening-Harvest stages.**

**Table 1: Summary of daily, daytime and nighttime O₃ Vₐ and flux during the O-W, G-F and R-H stages.**

| | | Over-Wintering | | | Green-Flowering | | | Ripening-Harvest | | |
|---|---|---|---|---|---|---|---|---|---|---|
| | | All | Day | Night | All | Day | Night | All | Day | Night |
| $V_d$ | Mean | 0.20 | 0.29 | 0.12 | 0.36 | 0.46 | 0.24 | 0.20 | 0.30 | 0.05 |
| $(cm\ s^{-1})$ | Standard Deviation | 0.28 | 0.30 | 0.24 | 0.37 | 0.41 | 0.28 | 0.25 | 0.26 | 0.14 |
| | Median | 0.12 | 0.24 | 0.07 | 0.28 | 0.38 | 0.17 | 0.15 | 0.26 | 0.07 |
| | 75% | 0.33 | 0.39 | 0.22 | 0.51 | 0.61 | 0.37 | 0.33 | 0.48 | 0.13 |
| | 25% | 0.01 | 0.11 | -0.02 | 0.11 | 0.20 | 0.06 | 0.05 | 0.12 | -0.01 |
| O₃ flux | Mean | -0.10 | -0.19 | -0.02 | -0.28 | -0.42 | -0.12 | -0.27 | -0.40 | -0.06 |
| $(\mu g\ m^{-2}\ s^{-1})$ | Standard Deviation | 0.18 | 0.22 | 0.06 | 0.45 | 0.51 | 0.27 | 0.37 | 0.41 | 0.12 |
| | Median | -0.03 | -0.11 | -0.01 | -0.10 | -0.26 | -0.02 | -0.12 | -0.27 | -0.04 |
| | 75% | 0.00 | -0.03 | 0.00 | -0.01 | -0.08 | -0.01 | -0.02 | -0.08 | 0.00 |
| | 25% | -0.11 | -0.28 | -0.03 | -0.36 | -0.53 | -0.10 | -0.40 | -0.62 | -0.09 |



## 3.4 O₃ deposition relation to environmental factors





To gain deeper insights into the responses of $O_3$ deposition (including stomatal and non-stomatal) to environmental factors in agricultural areas, stepwise MLR models were conducted to see which factors potentially played more important roles in $O_3$ deposition at GC during daytime and nighttime, respectively. As shown in Table 2, RH, $u_*$, soil VWC and LAI were identified as significant environmental factors in explaining daytime $O_3$ deposition changes during the entire observation period. Additionally, the coefficients of determination ($R^2$) of the linear model between all environmental factors and $V_d$ was 0.46, implying that these meteorological and plant growth related factors could explain approximately 46% of the variance of daytime $O_3$ deposition, while $R^2$ was only slightly lower (0.43) with the four selected factors. Distinct key environmental factors for $O_3$ deposition were identified for different wheat growth stages, while LAI was only among the most important factors during the G-F stage (Table 2), confirming the significant effect of crop on $O_3$ deposition during its most vigorous growing stage. $O_3$ deposition was more sensitive to $u_*$, soil VWC and PAR during the O-W stage, while more affected by $T_{Air}$ and $u_*$ in the R-H stage. During nighttime, the most significant influencing factors for $O_3$ deposition were $T_{Air}$, $u_*$, $T_{Soil}$ and soil VWC for the whole observation period (Table 2). However, temperatures ($T_{Air}$ and $T_{Soil}$) were not selected as the key factors when the stepwise MLR was separately done for the three growth stages. Compared with the $R^2$ of daytime MLR model, meteorological and soil conditions could explain more variance of the nighttime $O_3$ deposition (54%), implying that nighttime $O_3$ deposition processes were less complicated than daytime ones.

**Table 2: Results of the MLR models for the O-W, G-F and R-H stages. Daytime MLR models represent multiple linear regression between daily average environmental variables and daytime $O_3$ $V_d$, while nighttime models represent nighttime environmental variables and $V_d$. The selected MLR models refer to the stepwise MLR model based on AIC statistics. Bold numbers denote those with p-value < 0.05.**

| | Whole Period | | | Over-Wintering | | | Green-Flowering | | | Ripening-Harvest | | |
|---|---|---|---|---|---|---|---|---|---|---|---|---|
| | MLR | Selected MLR | | MLR | Selected MLR | | MLR | Selected MLR | | MLR | Selected MLR | |
| | Coef. | Coef. | Std. Err. | Coef. | Coef. | Std. Err. | Coef. | Coef. | Std. Err. | Coef. | Coef. | Std. Err. |
| | | | | | Daytime | | | | | | | |
| $T_{Air}$ | -0.09 | | | 0.13 | | | -0.08 | | | 0.01 | **0.09** | **0.02** |
| RH | -0.08 | **-0.06** | **0.03** | 0.04 | | | -0.07 | -0.08 | 0.04 | 0.07 | | |
| $u_*$ | 0.04 | 0.05 | 0.03 | 0.04 | **0.09** | **0.02** | 0.08 | 0.08 | 0.04 | -0.07 | **-0.06** | **0.02** |
| VPD | -0.06 | | | -0.05 | | | -0.09 | | | 0.11 | | |
| $T_{Soil}$ | 0.11 | | | 0.00 | | | 0.24 | | | -0.21 | | |
| soil VWC | 0.10 | **0.10** | **0.04** | -0.01 | -0.05 | 0.02 | 0.14 | 0.07 | 0.06 | -0.36 | | |
| PAR | 0.01 | | | 0.04 | -0.04 | 0.04 | 0.05 | | | 0.03 | | |
| LAI | 0.19 | 0.07 | 0.05 | -0.06 | | | 0.30 | 0.08 | 0.06 | 0.60 | | |
| FPAR | -0.14 | | | -0.05 | | | -0.34 | | | -0.45 | | |
| $R^2$ | 0.46 | 0.43 | | | 0.93 | | 0.53 | 0.47 | | 0.64 | 0.56 | |



| | | | | | | | Nighttime | | | | | |
|---|---|---|---|---|---|---|---|---|---|---|---|---|
| $T_{Air}$ | -0.24 | **-0.20** | **0.08** | 0.05 | | | -0.15 | | | 0.12 | | |
| RH | -0.07 | | | 0.08 | | | -0.11 | | | -0.08 | -0.03 | 0.02 |
| $u_*$ | 0.02 | **0.03** | **0.02** | -0.07 | | | -0.01 | | | 0.02 | **0.03** | **0.02** |
| VPD | -0.01 | | | 0.08 | | | -0.02 | | | -0.11 | | |
| $T_{Soil}$ | 0.20 | **0.18** | **0.06** | 0.20 | | | 0.12 | | | -0.11 | | |
| soil VWC | 0.12 | **0.12** | **0.02** | 0.15 | **0.07** | **0.02** | 0.11 | **0.12** | **0.02** | -0.10 | -0.03 | 0.02 |
| $R^2$ | 0.54 | 0.49 | | 0.99 | 0.71 | | 0.57 | 0.39 | | 0.58 | 0.52 | |

Further, we explored how the selected key factors controlled the temporal variability of $V_d$ during the three wheat growth stages (Figures 6-8). In general, the responses of both daytime and nighttime $V_d$ to the meteorological factors were consistent

throughout the entire wheat growth season (Figures 6 and 8). A clear negative impact of RH on $V_d$ was observed at GC for daytime and nighttime (Figure 6a and 8b). Due to its negative correlation to RH (as was shown in Eq.6), high VPD was conducive to high nighttime $V_d$ (Figure 8c). Similar response of $V_d$ to humidity were observed over wheat and maize canopy in the NCP (Zhu et al., 2014; Zhu et al., 2015), while $O_3$ deposition into a boreal forest revealed strong positive correlation with RH (Rannik et al., 2012), which might be attributed from the differences in plant species and growth environment. For

plants, the response of stomata to the changes of humidity is largely dependent on the plant species, plant water stress and humidity condition (Camacho et al., 1974; Rawson et al., 1977; Fanourakis et al., 2020). For example, with the increase of VPD, stomatal conductance of a tropical rainforest gradually increased (Mendes and Marenco, 2017), while the leaf diffusion resistance of sesame also increased markedly (Camacho et al., 1974). Thus, the similar change of humidity might have opposite effect on $O_3$ deposition over different canopies. Besides, $O_3$ deposition processes also varied with the humidity.

Under dry condition (RH < 60%), $O_3$ deposition over crop canopy is significantly controlled by stomatal uptake; above 60-70% RH, a thin liquid film on leaf surface will block dry deposition but enhance the aqueous reactions of $O_3$, thus the contribution of non-stomatal deposition is higher and more variable (Coyle et al., 2009; Lamaud et al., 2009). Strong turbulence (represented by elevated $u_*$) can transport $O_3$ more efficiently to the surface (Cape et al., 2009), thus $V_d$ almost linearly increased with $u_*$ (Figure 6b and 8d). The sensitivity of $V_d$ to $u_*$ was also affected by LAI, with daytime $V_d$ under

similar levels of $u_*$ being significantly higher under higher LAI (Figure 7a and Table S3). High LAI indicates dense vegetation coverage and potential large stomatal conductance, which can provide more active (stomatal and cuticular) areas for the uptake of $O_3$, further promoting $O_3$ deposition. PAR had a positive effect on $O_3$ deposition during the observation period (Figure 6c). On the one side, increasing PAR induces automatic leaf stomatal opening thereby determining stomatal conductance and net photosynthesis (Yu et al., 2004), affecting the stomatal $O_3$ uptake (Tong et al., 2015). On the other side,

PAR (also reflecting radiation intensity) affects $O_3$ photochemistry directly by accelerating atmospheric photolysis reactions both above and within the canopy and indirectly by influencing the emission of biogenic VOCs (Yang et al., 2021; Van




Meeningen et al., 2017; Yuan et al., 2016), thus disturbing the distributions of $O_3$ and its precursors and contributing non-stomatal $O_3$ fluxes with surface processes (Fares et al., 2008; Cape et al., 2009). Positive dependences of $V_d$ on $u_*$ and PAR were observed over other crop fields (such as wheat, maize, and potato) (Coyle et al., 2009; Zhu et al., 2014; Zhu et al., 2015). In addition, both $T_{Air}$ and $T_{Soil}$ exhibited weak relationships with nighttime $V_d$ (Figures 8a, e), which were different from the reported positive correlations between temperature and $V_d$ (Coyle et al., 2009; Rannik et al., 2012). These imply the variability and complexity of $O_3$ deposition affected by the combined influences of various environmental factors.

During the O-W and R-H stage, soil VWC was at relatively lower levels. During the G-F stage, soil VWC reached beyond 0.30% and $V_d$ rose significantly with rising soil VWC (Figure 6d and 8f). Although soil moisture blocks the diffusion of $O_3$ in soil and reduces reactive spaces for $O_3$ absorption, suppressing $O_3$ soil deposition (Stella et al., 2011), it can also promote total $O_3$ deposition through several indirect pathways. From the plant physiological aspect, stomatal conductance and plant net photosynthesis are both promoted by higher soil VWC (Otu-Larbi et al., 2021; Anav et al., 2018; Medlyn et al., 2011; Jarvis et al., 1997; Ball et al., 1987). Stomatal conductance revealed overshoots after the watering of dry soil, accordingly, significantly increased transpiration and photosynthesis of crops or vegetation were detected (Wu et al., 2021; Reich et al., 2018; Ramírez et al., 2018; Rawson and Clarke, 1988; Popescu, 1967). This effect was reflected at GC by the more obvious response of $V_d$ to soil VWC changes under higher LAI (Figure 7b and Table S3), confirming again that $O_3$ deposition during the G-F stage was mainly driven by stomatal deposition, rather than soil deposition. Consequently, soil VWC revealed positive coefficients in the MLR models for the G-F period at GC. Our result is qualitatively consistent with the observation based estimation of stomatal and soil $O_3$ deposition relative contributions over a wheat canopy in Nanjing city, China, which accounted for 41.2% and 15.4%, respectively (Xu et al., 2018).

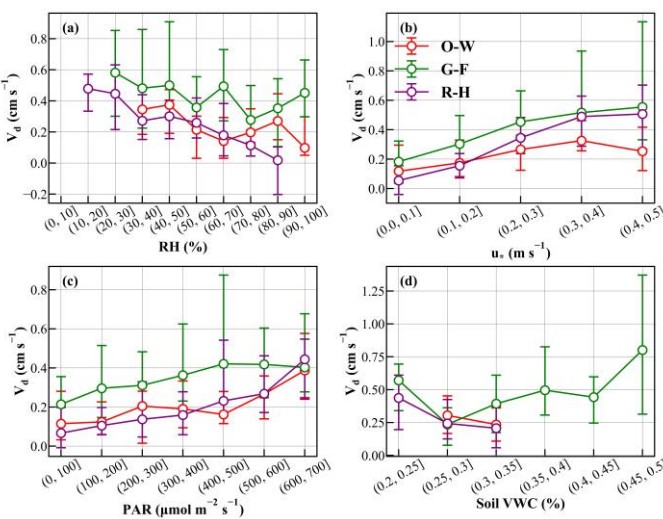

**Figure 6: Dependences of daytime $O_3$ $V_d$ on (a) RH, (b) $u_*$, (c) PAR and (d) soil VWC during the O-W, G-F and R-H stages. Medians of 30-min $V_d$ with quartiles are presented.**



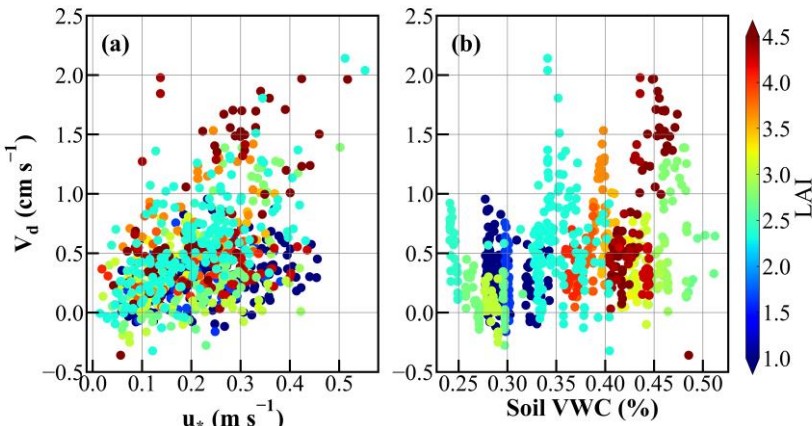

Figure 7: The variation of daytime $V_d$ with (a) $u_*$, (b) soil VWC under changing LAI during the G-F stage.

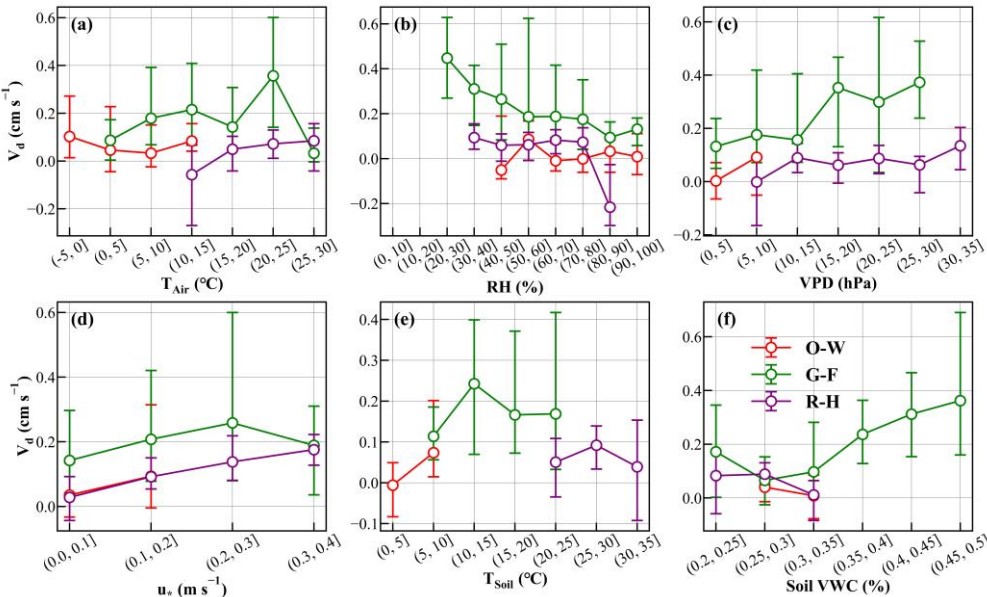

Figure 8: Dependences of $O_3$ $V_d$ on (a) $T_{Air}$, (b) RH, (c) VPD, (d) $u_*$, (e) $T_{Soil}$ and (f) soil VWC in the nighttime during the O-W, G-F and R-H stages. Medians of 30-min $V_d$ with quartiles are presented.

Overall, soil VWC at GC kept low expect for four abrupt increasing events that occurred during the G-F stage, with the highest soil VWC reaching 0.55% (Figure 2c). The three most prominent episodes induced by farm field irrigation were more thoroughly investigated to uncover how soil moisture affected $O_3$ deposition at GC. Interestingly, simultaneous increments in $V_d$ were detected upon the increase of soil VWC, with $V_d$ being distinctly elevated within days afterwards (Figure 9). Soil VWC increased from 0.32% to 0.51% at noontime on 9 April, and $V_d$ rapidly rose from 0.16 to 0.64 cm s$^{-1}$ at the same time. Dramatic increments in both averaged nighttime and daytime $V_d$ were detected on the following days. During



nighttime, $V_d$ reached an average of 0.30 cm s$^{-1}$, significantly higher than the average daytime level (0.18 cm s$^{-1}$) on 9 April, and a maximum of 0.76 cm s$^{-1}$, which also exceeded the maximum of 0.64 cm s$^{-1}$ observed during the daytime of 9 April (Figure 9a). The dramatic rise in $V_d$ on 10 April resulted a 317% increase in daytime $O_3$ deposition flux (from 0.12 μg m$^{-2}$ s$^{-1}$ to 0.50 μg m$^{-2}$ s$^{-1}$), with only a small change in daytime $O_3$ concentration from 9 April (41.4 μg m$^{-3}$) to 10 April (48.2 μg m$^{-3}$, Figure S4a). In addition, drastic elevations in $V_d$ during the night and morning periods were also observed following other

episodes of sudden increases in soil VWC (Figure 9b, c). Similar enhancements and disrupted daily cycles of $O_3$ deposition were also observed over the canopy of a pine forest during rainfall events (Altimir et al., 2006).

Considering the direct effect of soil moisture on plant physiology, the temporal variations of $CO_2$ and $H_2O$ fluxes were examined to characterize changes in transpiration and photosynthesis of wheat affected by the abrupt increases of soil water contents. As shown in Figure 10, both $CO_2$ and $H_2O$ fluxes exhibited obvious increases on the days following the soil VWC

increments. The daily peaks of $H_2O$ fluxes increased from 0.08 to 0.13 g m$^{-2}$ s$^{-1}$ between 9 and 10 April and from 0.12 to 0.19 g m$^{-2}$ s$^{-1}$ between 28 and 29 April, respectively, while daily averaged $CO_2$ fluxes before and after the abrupt increasing events rose from 0.28 mg m$^{-2}$ s$^{-1}$ to 0.55 mg m$^{-2}$ s$^{-1}$ and 0.51 mg m$^{-2}$ s$^{-1}$ to 0.55 mg m$^{-2}$ s$^{-1}$, respectively (Figure 10a, b). Subsequently, $CO_2$ and $H_2O$ fluxes, as well as $V_d$ of $O_3$ exhibited declines with the slow loss in soil moisture (Figure 9 and 10). This indicates that transpiration and photosynthesis of wheat were sharply enhanced after soil water contents increased,

leading to larger leaf stomatal conductance and strengthening $O_3$ stomatal uptake. These results were consistent with those obtained in previous conditional control experiments and field observations (Wu et al., 2021; Reich et al., 2018; Ramírez et al., 2018; Rawson and Clarke, 1988; Popescu, 1967). In addition, moist soil can extend the time window of wheat leaves' stomatal opening, both in the hours after sunset and before dawn (Schoppach et al., 2020; Ramírez et al., 2018). Stomata can even stay open during nighttime after precipitation or irrigation events (Kobayashi et al., 2007; Rawson and Clarke, 1988).

During the irrigation induced high soil VWC episodes, positive $H_2O$ fluxes were also observed at GC during the night, such as on 10 April and 28 April (Figure 10a, b), implying that wheat transpiration might not stop over the night and that leaf stomata might have not completely closed, continuing to uptake $O_3$ at night, significantly enhancing nocturnal $O_3$ deposition.

Additionally, the high nighttime $O_3$ deposition ($V_d$) phenomenon was always accompanied by positive water vapor fluxes and high NO concentrations, and occurred mainly after the rapid increase of soil VWC (Figure S5). As shown in Figure 9,

high NO became more frequent at night during the high soil VWC events, and nighttime $V_d$ dramatically increased when NO fluctuated at obviously high levels and nighttime $O_3$ concentration was still at low level (Figure S4). This might be attributable to the fact that soil NO emissions were promoted by the watering process, as soil water content is a decisive factor for the transformation and emission of reactive nitrogen within soils (Schindlbacher et al., 2004; Ghude et al., 2010; Kim et al., 2012; Weber et al., 2015; Zörner et al., 2016). Enhanced nighttime soil NO emissions may inevitably cause

stronger NO titration with $O_3$ within wheat canopy, facilitating the non-stomatal $O_3$ deposition at night.



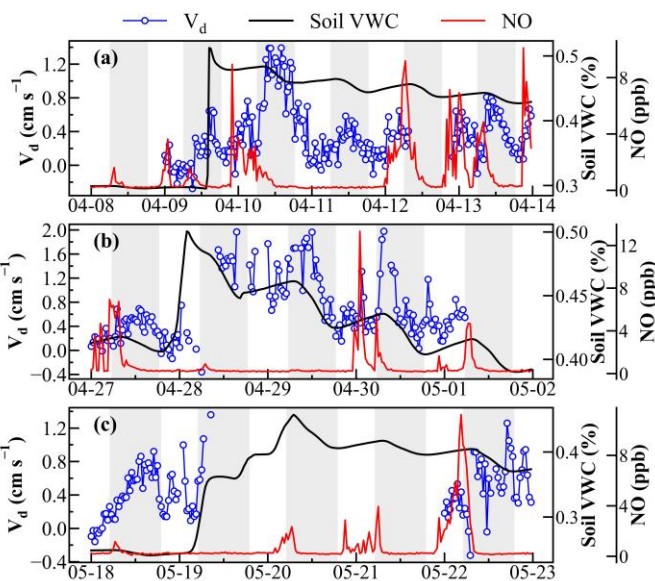

**Figure 9: Variations of $O_3$ $V_d$ (blue circle lines), soil VWC (black lines) and NO concentration (red lines) during 8-13 April (a), 27 April-1 May (b) and 18-22 May (c). The shades represent the daytime.**

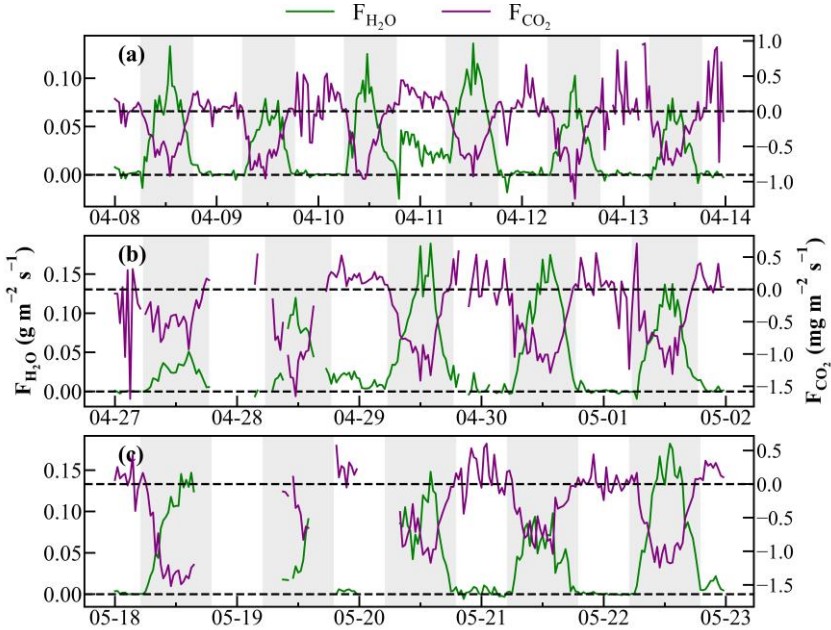

**Figure 10: Variations of $H_2O$ flux ($F_{H_2O}$, green lines) and $CO_2$ flux ($F_{CO_2}$, purple lines) during (a) 8-13 April, (b) 27 April-1 May and (c) 18-22 May, with shades representing daytime hours.**

In summary, both daytime and nighttime $O_3$ deposition fluxes and $V_d$ were significantly affected by environmental conditions, through stomatal and nonstomatal pathways, with crop growth playing a critical role. Abrupt increments in soil





moisture induced dramatic changes in $V_d$, which not only altered the diurnal cycle of $O_3$ deposition, but also caused large

fluctuations in averaged $O_3$ deposition flux on longer timescales.

**4 Conclusions and implications**

    In this study, we developed a relaxed eddy accumulation (REA) $O_3$ flux measurement system, verified its reliability, and conducted measurements of $O_3$ deposition using this newly developed REA system over the wheat canopy at a polluted agricultural site (GC) in the NCP during the main wheat growing season. We also collected ancillary data related to $O_3$

deposition and tried to make an integrated analysis. The observed $O_3$ deposition flux and velocity over the wheat fields at GC averaged $-0.25 \pm 0.39$ $\mu g\ m^{-2}\ s^{-1}$ and $0.29 \pm 0.33$ $cm\ s^{-1}$, respectively. The diurnal cycle of $V_d$ was controlled by the crop stomatal opening during the day. Daytime $V_d$ ($0.40 \pm 0.38$ $cm\ s^{-1}$) was obviously higher than nighttime one ($0.17 \pm 0.26$ $cm$ $s^{-1}$). $V_d$ played a decisive role in the diel pattern of $O_3$ deposition flux, while $O_3$ concentration determined the flux variability on the longer timescales. The temporal changes of $V_d$ were synchronous with the evolutions of LAI and wheat growth,

suggesting the determining effect of crops on $O_3$ uptake and the predominant contribution of stomatal uptake to the overall $O_3$ deposition. During the wheat growth season, RH, $u_*$, soil VWC and LAI were identified as the most significant factors in explaining the changes of $O_3$ deposition in the daytime, while $u_*$ and soil VWC were more important in the nighttime. $V_d$ exhibited significant increases with the decrease of RH and increases of $u_*$, PAR and soil VWC, which were enhanced by higher LAI. With the rapid increases of soil VWC after strong precipitation or irrigation events, stomatal conductance

increased and stomatal opening extended to nighttime hours, enhancing transpiration and photosynthesis of wheat, which remarkably strengthened $O_3$ stomatal uptake during daytime and nighttime. Meanwhile, soil NO emission might have also been strengthened under moist soil conditions, which facilitated NO titration of $O_3$ within the canopy and enhanced non-stomatal $O_3$ removals at night. Therefore, drastically increasing soil moisture led to the dramatic and simultaneous increments in $V_d$.

Observations have proved the dominant effects of crop growth on $O_3$ deposition processes over wheat fields during its growth season, whereas the actual relative contributions of crops to $O_3$ deposition through different pathways need to be further quantified. On the other hand, the influences of crops on $O_3$ deposition through stomatal uptake or surface removal have been investigated in extensive studies (Ainsworth, 2017; Aunan et al., 2000; Bender and Weigel, 2011; Biswas et al., 2008; Epa, 2013; Felzer et al., 2005; Harmens et al., 2018; Piikki et al., 2008). How much impact will $O_3$ deposition have on

crop growth and yield under the currently severe $O_3$ pollution with significant upward trends in the NCP remains an unsolved issue. Although many researches have assessed the crop yield loss caused by $O_3$ pollution based on exposure-response functions (Feng et al., 2019a; Hu et al., 2020; Feng et al., 2020), the concentration-based evaluation was less related to the actual exposure than deposition flux (Zhu et al., 2015). Therefore, more accurate quantification of agricultural impacts induced by $O_3$ (based on $O_3$ deposition fluxes) are required in future studies.



During the wheat growth season, the temporal variation of $O_3$ deposition velocity fluctuated greatly with changes in environmental conditions, and the dominant factors determining the $V_d$ variability varied with the growth stages of wheat. Meanwhile, the key influencing factors and their effects on $O_3$ deposition varied with different canopy and ground surface. Beside environmental conditions, agricultural activities also significantly affected $O_3$ deposition (Mészáros et al., 2009). Thus, using a unanimous $O_3$ deposition parameterization scheme on different plant canopies/surfaces or throughout distinct

growth stages might cause large errors in simulation results, explaining the large discrepancies between modelled and observed $O_3$ dry deposition fluxes in the growth seasons (Clifton et al., 2020b; Hardacre et al., 2015). More $O_3$ deposition observations over different land surfaces are definitely needed in the future, both facilitating the $O_3$ dry deposition mechanism exploration and model optimization.

    Finally, abrupt increases in soil moisture during strong precipitation and irrigation resulted in dramatic and overall

increases in $V_d$ over the wheat fields, and extremely high nighttime $V_d$ was observed, affecting the total $O_3$ deposition. Impacted by climate warming, more frequent irrigation in agricultural areas might promote $O_3$ deposition following the drier and hotter days, when the photochemical production of $O_3$ is greatly enhanced and $O_3$ concentration is at a high level (Yuan et al., 2016; Lu et al., 2019). Meanwhile, increasing temperature promotes the emissions of soil NO and biogenic VOCs (Ma et al., 2019; Lu et al., 2020), further facilitating $O_3$ production and deposition. Correspondingly, the feedback of these

climate-related impacts on the vegetation growth is worth further study. On the other hand, the processes of stomatal opening and transpiration have been commonly assumed to occur during daytime and stop during the night. However, more and more studies have showed the non-negligible effects of nocturnal unclosed stomatal and transpiration for a wide range of plant species (Kukal and Irmak, 2022; Schoppach et al., 2020; Tamang et al., 2019; Ramírez et al., 2018; Hoshika et al., 2018), and the viable breeding target on increasing pre-dawn circadian control would be set to alleviate the adverse effect of

increasing drought under climate warming (Schoppach et al., 2020). With the significant increasing $O_3$ levels in China, nighttime $O_3$ concentration and exposure also exhibited upward trend in recent years (Agathokleous et al., 2023; He et al., 2022). How nocturnal plant activities and rising nighttime $O_3$ concentration interact with each other is worth deeper investigation.

**Data availability**

The data used in this study are available from the corresponding authors upon request.

**Author contributions**

XY and WX designed the experiment and XB led the research. XY made the $O_3$ deposition measurement with the help of WL, WX, ZG, XB and JM. JJ, ZL, SX, HR and GS were responsible for the EC flux measurement. XY analyzed the data and wrote the paper with help of WL, WX, XB.



## Competing interest

The authors have no competing interests to declare.

## Acknowledgments

This work is supported by the National Natural Science Foundation of China (42175127 and 41875159), Beijing Natural Science Foundation (8222078), CAMS projects (2023Z012 and 2020KJ003).

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
