# Peer review of "Ozone deposition measurements over wheat fields in the North China Plain: variability and related factors of deposition flux and velocity"

_EGUsphere, 2024_

## Author Comment (AC1)

**Response to comments by Anonymous referee #2**

The manuscript presents a detailed study on ozone ($O_3$) deposition over wheat fields in the North China Plain. The authors have employed a newly developed relaxed eddy accumulation (REA) system to measure $O_3$ deposition flux and velocity during the 2023 wheat growing season. The study explores the variability of $O_3$ deposition and its influencing factors, emphasizing the importance of crop growth and environmental conditions.

Overall, the study addresses a significant gap in the measurement and understanding of $O_3$ deposition in agricultural settings in China, a region experiencing increasing $O_3$ pollution. The use of the REA system is novel and relevant for capturing detailed deposition metrics. The results are comprehensive, showing the variation in $O_3$ deposition flux and velocity across different times of the day and growth stages of wheat. The discussion on the predominant role of stomatal uptake and the influence of environmental factors like relative humidity, soil water content, and friction velocity is noteworthy. I would hope the following points are considered or addressed:

**Response**: We appreciate your time spent reviewing our manuscript and are grateful for your constructive comments and suggestions. We have revised the manuscript according to your suggestions, and made point-to-point responses, with changes in the manuscript highlighted in yellow color.

Line 52: Are there any variations?

**Response**: Yes, both the relative contributions of non-stomatal and stomatal $O_3$ deposition varied with clear diurnal cycles and changed with crop growth stage. Stomatal deposition was more pronounced during mid-day and during the most rigorous growth stage of plant. We added the variability of stomatal $O_3$ deposition in Lines 49-52:

"For example, the fraction of diurnal maximum stomatal $O_3$ deposition over boreal forests ranged from 56 to 74% (Rannik et al., 2012), while only accounting for 31.2% in a wheat field (Xu et al., 2018), with both of them peaking at mid-day during the most rigorous growth stage of vegetations (Xu et al., 2018; Rannik et al., 2012)."

Line 65: Meters?

**Response**: No, "2~4" refers to the number of height layers at which the FG methods needs to make simultaneous $O_3$ concentration measurements in order to obtain the vertical gradient of $O_3$ levels. This part was deleted from the revised manuscript following suggestions of Reviewer #1.

Line 91: What do you mean by polluted agricultural areas?

**Response**: The Gucheng site is located in a typical agricultural area of the North China Plain, that is frequently under severe regional air pollution. For example, the overall average MDA8 $O_3$ in the warm season (April - September) during the 2006-2019 at the site was 64 $\pm$7.4 ppb, which was close to the Ambient Air

Quality Standards (standard code: GB3095-2012) of 75 ppb. The largest exceedance frequency of $O_3$ could exceed 50% in 2015, with averaged MDA8 $O_3$ on exceedance days reaching 102.1 ppb, indicating remarkable $O_3$ pollution exposure at Gucheng site (Zhang et al., 2022). We rephrased this sentence as follows:

Lines 89-91: "Observations at the site have revealed good regional representativeness of the agricultural areas in the NCP, that is heavily impacted by the severe regional air pollution (Lin et al., 2009; Xu et al., 2019; Kuang et al., 2020; Zhang et al., 2022a, b)."

Reference:

Zhang, X., Xu, W., Zhang, G., Lin, W., Zhao, H., Ren, S., Zhou, G., Chen, J., and Xu, X.: First long-term surface ozone variations at an agricultural site in the North China Plain: Evolution under changing meteorology and emissions, Sci.Total Environ., 160520, https://doi.org/10.1016/j.scitotenv.2022.160520, 2022.

For the fast response solenoid valve, can you provide the response time of them, and how does it compare with wind speed data frequency?

**Response**: Thank you for the suggestions. The response time of the fast-response 3-way solenoid valve was within 10 ms, and the estimated residence time from the REA system inlet to the valves was 18 ms, thus the total delay time from the inlet to the individual sampling tubes was less than 10 ms. The temporal resolution of three wind components was 100 ms, which was much larger than 10 ms. Thus, the REA system could work at a sampling frequency of 10 Hz (100 ms) according to the instantaneous vertical wind speed. The following description of the system response times was added:

Lines 129-134: "The estimated residence time from system inlet to the valves was 18 ms, while the response time of the fast-response sampling valves was less than 10 ms, leading to total time delays from inlet to individual sampling tubes of below 10 ms, thus the REA system could work at a sample frequency of 10 Hz (100 ms). Total residence time of air samples from the tip of the inlet to the point of $O_3$ detection was about 10 s, which was much shorter that the lifetime of $O_3$ reacting with NO (Supplementary methods), suggesting the chemical reaction in the two channels could be neglected."

The flow rate of your inlet sample seems to be missing. Have you synchronized your sonic data and the ozone data considering the length of your inlet tubing? Please add this information.

**Response**: Thank you for the reminder. We added the flow rates of the system tubes in Figure 1. The temporal revolution of sonic data is 100 ms, which is 10-fold that of the estimated residence time from the inlet to the individual sampling tubes. Thus, we just synchronized the sonic data and $O_3$ data according to the PC time. We added the information in Section 2.2.2.

Lines 134-135: "The $O_3$ analyzers recorded 1-minute averaged $O_3$ concentrations, which were downloaded by the PC. $O_3$ data were synchronized with wind data as well as sample time according to the PC time."

[Figure]

**Figure 1. Schematic of the REA-O₃ flux system ($w > w_0$).**

Lines 144-145: This sentence reads a bit unclear, please rewrite.

**Response**: Thanks for your suggestion. We revised this sentence as "Using raw EC data, the REA $CO_2$, $H_2O$ and heat fluxes were calculated under $w_0$ in this study (0.05 m s⁻¹ for daytime and 0.01 m s⁻¹ for nighttime) and $w_0 = 0$, respectively, with a constant $b$ of 0.60 (Businger and Oncley, 1990)."

Fig. S1: Is it just the data from a randomly selected day? If yes, is it from the growing season?

**Response**: The flux data in Figure S1 are from the whole observation period (from 12 February to 18 June 2023). To avoid the misleading, we added the time of flux data in the discussion in Section 2.2.2 and the title of Figure S1.

Lines 157-158: "As shown in Figure S2, the two flux datasets revealed excellent correlations during the whole observation period, with a correlation coefficient close to 1, confirming the reliability and stability of the REA flux measurement system."

**"Figure S1.** The influences of wind dead-bands ($w_0$) on (a, d) REA-$H_2O$, (b, e) REA-$CO_2$ and (c, f) REA-heat fluxes during (a-c) daytime and (d-f) nighttime from 12 February to 18 June, driven by EC raw data with a constant $b = 0.60$. $F^*$ and $F$ represent the REA fluxes with and without $w_0$, respectively. The linear regressions and correlation coefficients (r) between $F^*$ and $F$ are inset in each figure, and n is the total number of the valid fluxes.**"**

How does the agreement look during the non-growing season when plant metabolism is low? Environmental variability (e.g., changes in turbulence, vegetation state) could influence the optimal $w_0$ value. Different stages of crop growth and varying meteorological conditions likely require different $w_0$ adjustments. A static $w_0$ value for all conditions is overly simplistic and likely inadequate for capturing the true dynamics of ozone deposition.

**Response**: Thank you for the comment. We regarded the $CO_2$ flux in the agricultural ecosystem as an indicator for plant metabolism, and further investigated the influence of crop growth on $O_3$ deposition. We added the variation of cropland $CO_2$ flux in Figure 6, and adjusted the discussion as follows:

Lines 273-282: "To investigate the influences of wheat growth on $O_3$ deposition, the characteristics of $O_3$ deposition were further examined in connection to the different growth stages. During the O-W stage, wheat was in dormancy and leaves had not begun to turn green (LAI < 0.5, Figure 6b), with $CO_2$ flux in the agricultural ecosystem closed to zero (Figure 6c). $V_d$ in the O-W stage barely changed, exhibiting a low average value of $0.20 \pm 0.28$ cm s$^{-1}$ and a median of 0.12 cm s$^{-1}$ (Table 1). Wheat grew vigorously in the G-F stage, with LAI and $CO_2$ assimilation flux exhibiting rapid increases until the early and late flowering stage, respectively, after which both of them gradually decreased (Figure 6b-c). $O_3$ deposition varied nearly in synchronization with LAI and wheat growth, with $V_d$ reaching a peak when cropland $CO_2$ assimilation was the highest during the G-F stage (Figure 6a), reaching highest daytime and nighttime averages of $0.46 \pm 0.41$ cm s$^{-1}$ and $0.24 \pm 0.28$ cm s$^{-1}$, respectively (Table 1). Afterwards, with the maturing of wheat and the aging of leaves in the R-H stage, $V_d$ quickly dropped back to a low average level of $0.20 \pm 0.25$ cm s$^{-1}$, similar to that observed in the O-W stage."

[Figure]

**Figure 6.** (a) $O_3$ $V_d$, (b) LAI and FPAR, (c) $CO_2$ flux ($F_{CO_2}$) in different wheat growing stages. The circles and error bars in (a) denote the weekly medians and quantiles of $V_d$, respectively. O-W, G-F and R-H represent Over-Wintering, Green-Flowering and Ripening-Harvest stages.

We agree on the notion that environmental variability could affect the choice of the optimal $w_0$ values, and that adopting a dynamic $w_0$ could reduce the impact of atmospheric environment variabilities on observed flux to certain extents (Grönholm et al., 2008; Nelson et al., 2017). In previously reported field experiments, the dynamic $w_0$ was mainly dependent on the deviation of vertical velocity (Grönholm et al., 2008), for example, $w_0 = \frac{\sigma_w}{0.35}$ in the REA system of Grelle and Keck (2021) for $H_2O$, $CO_2$, $CH_4$, $N_2O$ flux measurements. During our observation period, $u_*$ fluctuated in the range of 0.05-0.30 m s$^{-1}$, and exhibited no obvious seasonal changes with crop growth (Figure 2b), implying that the variabilities of vegetation state and other meteorological conditions at our site played minor roles in the determination of the optimal $w_0$ value.

The application of $w_0$ is used to promote the sampling of larger eddies that contribute more to gas fluxes, and filter out samples with small vertical displacements that have relatively small impacts on the overall flux (Grelle and Keck, 2021). A constant $w_0$ would set a uniform threshold of sampled eddies, which tend to have relatively greater effects on the flux regardless of the environmental condition. Therefore, using a constant $w_0$ value for all conditions is beneficial to the comparison of observed fluxes under different environmental condition. In our REA system, using two static $w_0$ leads to ~10% overestimation to the measured fluxes, which was comparable with the influence of the dynamic $w_0$ in Grelle and Keck (2021)'s REA system, indicating a minor difference in their effect on flux derivations. In addition, adopting a static $w_0$ can avoid sampling mistakes induced by miscalculations of the dynamic $w_0$ or by large disturbances of environmental factors during the measurement, reducing the flux measurement errors. Therefore, we adopted a constant $w_0$ in our REA system.

Reference:

Grelle, A. and Keck, H.: Affordable relaxed eddy accumulation system to measure fluxes of H2O, CO2, CH4 and N2O from ecosystems, Agricultural and Forest Meteorology, 307, 108514, https://doi.org/10.1016/j.agrformet.2021.108514, 2021.

Grönholm, T., Haapanala, S., Launiainen, S., Rinne, J., Vesala, T., and Rannik, Ü.: The dependence of the β coefficient of REA system with dynamic deadband on atmospheric conditions, Environ. Pollut., 152, 597-603, https://doi.org/10.1016/j.envpol.2007.06.071, 2008.

Nelson, A. J., Koloutsou-Vakakis, S., Rood, M. J., Myles, L., Lehmann, C., Bernacchi, C., Balasubramanian, S., Joo, E., Heuer, M., Vieira-Filho, M., and Lin, J.: Season-long ammonia flux measurements above fertilized corn in central Illinois, USA, using relaxed eddy accumulation, Agricultural and Forest Meteorology, 239, 202-212, https://doi.org/10.1016/j.agrformet.2017.03.010, 2017.

Have you checked if the fetch and footprint of your REA tower exclusively covered the crop field?

**Response**: Yes, the height of the flux tower was designed according to the result of the fetch and footprint analysis. The range of flux source region was about 400 m, which is covered by the crop field within the Gucheng observation station. We added this information in Section 2.2.2:

Lines 111-115: "A 3-D sonic anemometer (CSAT3, Campbell Scientific Inc., USA) was used for measuring the three wind components ($u$, $v$, $w$) at 10 Hz, which was amounted at the height of 4.5 m on an eddy covariance tower, located in the middle of cropland. The height of the flux tower was designed according to the result of the fetch and footprint analysis. The range of flux source region was about 400 m, which is covered by the crop field within the GC station."

Line 166: Specify the ancillary data you have obtained.

**Response**: Thanks for your suggestion, the ancillary data includes meteorology, plant growth indicators and trace gas measurement data. We revised this sentenced as "Ancillary data were obtained for further analysis, including meteorology data, soil parameters, plant growth indicators and $O_3$ related trace gas measurement data."

Section 2.4: Provide more details on how you have normalized your environmental parameters.

**Response**: Thank you for the suggestion, we added the normalization method in Section 2.4.

Lines 213-217: "The Z-Score normalization method was adopted according to the following equation:
$$x = (x_{observed} - x_{mean}) \div x_{standard\ deviation} , \hspace{3cm} Eq.\ (9)$$
where $x_{observed}$, $x_{mean}$ and $x_{standard\ deviation}$ are the observed parameters, its overall average and standard deviation, respectively."

Include the life stage information of the crops in the method section.
**Response**: Thanks for the suggestion. We moved the life stage of wheat in Section 3.3 to Section 2.3, and added the plant height in Section 2.3.

Lines 182-185: "According to the winter wheat phenology at GC (Table S1), its entire growth season could be divided into three stages: Over-Wintering (O-W, 13 February-5 March), Green-Flowering (G-F, 6 March-28 May) and Ripening-Harvest (R-H, 29 May-18 June). The wheat height increased from 6.0 cm during the O-W stage to 61.2 cm at the R-H stage."

---

## Author Comment (AC2)

**Response to comments by Anonymous referee #1**

**General comment:**

This paper presents ozone flux measurements over one growing season in 2023 of winter wheat crop in North China. The flux measurement technique used here has not been applied to ozone before and the paper includes a description of the experimental set up. Additional flux measurements of momentum, $CO_2$, $H_2O$ and sensible heat are used to interpret the ozone flux measurements and their quality. Other chemically relevant species such as $NO_x$ (NO and $NO_2$) are also measured. The ozone flux measurements are investigated in relation to the growth stage of the wheat crop, meteorological and soil conditions or events such as precipitation and irrigation events. Different deposition pathways (stomatal and non-stomatal) are considered, but the partitioning is not estimated. Since ozone flux measurements are not as ubiquitous and standardised as e.g. $CO_2$ flux measurements this paper makes a useful contribution and provides sufficient ancillary measurements for detailed investigation. Using a novel measurement technique that does not require fast response ozone analysers extends the variety of techniques that are available for ozone deposition studies.

**Response**: We sincerely thank the reviewer for the detailed evaluation of our manuscript and all the thoughtful and constructive comments. We have revised the manuscript according to the suggestions, and made point-to-point responses, with changes highlighted in yellow color in the revised manuscript.

The following changes should be made to improve the manuscript with respect to clarity, information content and thus value to the reader.

**Specific questions/issues:**

1 Introduction

The introduction could benefit from revision. Content in lines 61 to 67 could be shortened as a method is described that is not used in this study. Instead extend the introduction (including benefits and disadvantages) of the relaxed REA method. Include references to recent studies, e.g. REA over crop surfaces and review of the REA method.

**Response**: Thanks for your suggestion, we added the basic principles, advantages and requirements of the REA method and shortened the introductions of EC and FG methods.

"Currently, the eddy covariance (EC) method and flux-gradient (FG) approach are the most commonly used micrometeorological techniques for measuring $O_3$ vertical fluxes (Businger and Oncley, 1990; Altimir et al., 2006; Wu et al., 2015; Clifton et al., 2020). However, EC requires robust fast-response measurement instruments ($\geq 10$ Hz) (Hicks and Wesely, 1978; Muller et al., 2009), while the assumption of FG is dependent on surface roughness and the photochemical reactions of $O_3$ and its precursors (Raupach and Thom, 1981; Vilà-Guerau De Arellano and Duynkerke, 1992). These relatively high requirements have more or less limited the application of traditional micrometeorological methods to the measurements of $O_3$ flux. The relaxed eddy accumulation (REA) method is another important micrometeorological method for observing the air-surface exchange of interested substances over ecosystems (Desjardins, 1977; Businger and Oncley, 1990). REA overcomes the need for fast-response gas sensors, and is based on the same physical principle as EC without

introducing other uncertainties (Pattey et al., 1993). REA relies on the conditional sampling of air at a constant flow rate according to the instantaneous vertical velocity, which requires high-response sampling valves (~ 10 Hz). The air samples associated with updrafts and downdrafts are accumulated into two separate reservoirs and accurately measured with slow-response gas analyzers (Businger and Oncley, 1990). In addition, REA sampling systems are of low-cost, easily portable and simple to operate, which allows easy deployment at remote locations in forests, over croplands and grassland surfaces (Sarkar et al., 2020). Thus, REA methods have been widely applied in flux measurements of various species, such as biogenic VOCs (Mochizuki et al., 2014), reduced sulfur gases (Xu et al., 2002), HONO (Ren et al., 2011) and aerosols (Matsuda et al., 2015; Xu et al., 2021) above forest canopies, peroxyacetyl nitrate at a grassland site (Moravek et al., 2014) , $NH_3$ above fertilized corn (Nelson et al., 2017) and Hg at an urban site and over a boreal peatland (Osterwalder et al., 2016). To the best of our knowledge, the REA method has not been applied in $O_3$ deposition flux measurements so far."

2 Observation and method

2.1 Site description

Add information on the characteristics and thus suitability (e.g. fetch and footprint, topography, closest sites of e.g. anthropogenic $NO_x$ emissions) of the site for ecosystem flux measurements, adding references if available. The referenced figure 1a in Zhang et al, 2022b is not sufficient for this as a scale is missing.

**Response**: Thank you for the reminder, we apologize for the wrong citation here and changed (Zhang et al, 2022b) to (Zhang et al, 2022a). The surrounding topography and landcover of the GC site and the tropospheric $NO_2$ columns over the North China Plain were presented in Figure 1 in Zhang et al., 2022a. It can be seen from Figure 1a that the GC site is surrounded by farmlands with stranded villages, a national road to its west and highway connecting Beijing and Shijiazhuang further away to its east.

[Figure]

Fig. 1. The satellite map of GC site (left map) and the distribution of averaged tropospheric $NO_2$ columns in the warm seasons during 2006-2019 (right map)

We added the topography information of the surrounding areas to Section 2.1, the fetch and footprint of flux measurement in Section 2.2.2, and the satellite map of Gucheng station with a scale in Figure S1.

Lines 86-88: "The site is surrounded mainly by irrigated high-yield agricultural lands with small villages, and a highway connecting Beijing and Shijiazhuang 7 km to the west of the site (Figure S1)."

Lines 111-115: "A 3-D sonic anemometer (CSAT3, Campbell Scientific Inc., USA) was used for measuring the three wind components ($u$, $v$, $w$) at 10 Hz, which was amounted at the height of 4.5 m on an eddy covariance tower, located in the middle of cropland. The height of the flux tower was designed according to the result of the fetch and footprint analysis. The range of flux source region was about 400 m, which is covered by the crop field within the GC station."

[Figure]

Figure S1: The satellite map of Gucheng site.

Reference:

Zhang, X., Xu, W., Zhang, G., Lin, W., Zhao, H., Ren, S., Zhou, G., Chen, J., and Xu, X.: First long-term surface ozone variations at an agricultural site in the North China Plain: Evolution under changing meteorology and emissions, Sci.Total Environ., 160520, https://doi.org/10.1016/j.scitotenv.2022.160520, 2022a.

2.2. Relaxed eddy accumulation (REA) technique

2.2.1. Add general reference on eddy covariance technique and its variant, relaxed eddy accumulation.

**Response**: Thank you for the suggestion, we added references on eddy covariance (Desjardins, 1977) and relaxed eddy accumulation (Businger and Oncley, 1990), respectively.

Lines 95-97: "In the REA-$O_3$ flux system, conditional sampling is conducted according to the direction of vertical wind ($w$), which separates sampled air into updraft and downdraft reservoirs at a constant flow rate (Desjardins, 1977; Businger and Oncley, 1990)."

2.2.2 Line 112, height of mounting (4.5 m) is provided, also add information somewhere in the text on the maximum height of the crop.

**Response**: Thanks, we added the height of wheat during the measurement period to Section 2.3.

Lines 181-185: "Measurements of O₃ flux were conducted during the main wheat growing season in 2023, from the late of dormancy stage (13 February 2023) to wheat harvest (18 June 2023). According to the winter wheat phenology at GC (Table S1), its entire growth season could be divided into three stages: Over-Wintering (O-W, 13 February-5 March), Green-Flowering (G-F, 6 March-28 May) and Ripening-Harvest (R-H, 29 May-18 June). The wheat height increased from 6.0 cm during the O-W stage to 61.2 cm at the R-H stage."

Figure 1 and section 2.2.2. More technical detail will be useful to include in Figure 1 and/or the text. Materials used should be fully described: type/model, manufacturer of e.g. particle filters, valves, pumps, air compressor. Also, length of tubing should be provided for the inlet section (entry point up to ozone analyser). The sample flows of the ozone analysers will provide salient information. Residence time from system inlet to 3-way valves are given as 18 ms, however the residence time within the updraft/downdraft tube sections is also relevant to specify. Is this residence time constant or depending on the up/downdraft switch frequency? Total residence time from tip of the inlet to point of ozone detection is important as ozone is a reactive species and in a polluted environment such as this study, the sampled air will also include ozone reactants (e.g. up to 4 ppb NO were measured, see Fig S5). This potential effect should be considered and quantified as a measurement uncertainty. Please add information on the zero air, is it only ozone free air, or is it also scrubbed for other trace species (e.g. NOx), as well as possible drying. If the air is dry, it would be useful to include observations on whether the switching between ambient humid air and zero dry air causes effects on the ozone concentration measurements.

**Response**: Thank you for the suggestions. Accordingly, we have made the following modifications.

(1) We added the models and manufacturers of particle filters, fast-response sampling valves, sampling pumps, zero air supply and air compressor in Lines 117-128:

"The wind signals were processed by a program written in Python, which also sent switch command to two fast-response 3-way solenoid valves (LVM105R-5C, Sintered Metal Company, Japan) according to the vertical wind direction. Based on the direction of instantaneous vertical winds, sample air was drawn alternatively through updraft or downdraft sample tubes (1/4" OD Teflon) wrapped in aluminum foil and were analyzed by the two UV photometric O₃ analyzers (TE 49i, Thermo Fisher Scientific Inc., USA) installed at the ends of updraft and downdraft channels, respectively. Coarse particulate matter was filtered out of air samples using two particle filters (47 mm single stage filter assembly, Savillex, LCC., USA) before entering the O₃ analyzers (Figure 1). To ensure the stability of airflow in the O₃ analyzers and sampling system, zero air was supplied to the channel that was not sampling ambient air. The zero air was generated by an external air compressor (M104, Gast Manufacturing Inc., USA) and a zero-air generator (Model 111, Thermo Fisher Scientific Inc., USA), which removes O₃, NO, NO₂, CO and hydrocarbons from ambient air, but not water vapor. In addition, both sampling tubes were bypassed through a piston pump (617CD22, Gardner Denver Thomas Inc., USA), in order to increase the inlet sample flow and avoid the axial mixing before the solenoid valves."

(2) We added the flow rates of the system, the inner diameters and lengths of sample tubes in Figure 1.

[Figure]

**Figure 1: Schematic of the REA-O₃ flux system ($w > w_0$).**

(3) We added the response time (< 10 ms) of sampling valves and the estimated total residence time (< 10 s) of air samples from the inlet to O₃ analyzers in Lines 129-133:

"The estimated residence time from system inlet to the valves was 18 ms, while the response time of the fast-response sampling valves was less than 10 ms, leading to the total delay time from inlet to individual sample tubes below 10 ms, thus the REA system could work at a sample frequency of 10 Hz (100 ms). Total residence time of air sample from tip of the inlet to the point of O₃ detection was about 10 s, which was much shorter that the lifetime of O₃ reacting with NO (Supplementary methods), suggesting the chemical reaction in the two channels could be neglected."

The residence time is dependent on the sampling flow rate and tube length, not sampling frequency. During the measurement period, sampling flow rate of the REA system kept stable, and we also calibrated the sampling rate every month to avoid large shifts. Thus, the total residence time remained mostly the same over the observation period. We agree that the total residence time is an important parameter that will affect the measurement of O₃ concentration and flux. The sampling frequency of the REA system was set to 10 Hz following that of the vertical wind velocity ($w$), which quickly fluctuated around $w = 0$ most of the time. Thus, air sample was separated into the updraft or downdraft channels within 10 ms, and then synchronously pumped into the O₃ analyzers. The length, material and condition of two channels were same, and both of them were treated to avoid sunlight. Synchronized multipoint calibrations of the two channels were also conducted monthly. Thus, the physical and chemical reaction condition of sampled air in the two channels were assumed to be consistent. On the other hand, zero air (without SO₂, O₃, NO, NO₂, CO and hydrocarbons) was supplied to the channel that was not sampling ambient air, and mixed with the sampled air, thus gas concentrations in the two channels were distinctly lower than the ambient concentration.

For gases and particulate matter with low deposition velocity, the differences of gas concentration between updraft and downdraft should be lower than 1% at most conditions (Hicks and Mcmillen, 1984). The $O_3$ in the two shaded channels mainly reacted with NO, which would occur in timescales of a few minutes in agricultural fields (Zhang et al., 2024). Since there was no simultaneous observation of NO concentrations in the two channels, we estimated the lifetime ($\tau_{chem}$) of $O_3$ reacting with NO using ambient NO concentrations, which should be the lower limit of the actual $\tau_{chem}$ in the two channels. During the observation period, $\tau_{chem}$ of $O_3$ in the atmosphere ranged from 141 s to 31559 s, and was at least 10-fold that of the total residence time of the REA system. Therefore, the difference of the chemical $O_3$ consumption in the two channels would be minor, and might have a marginal effect on flux measurements.

We added the evaluation of the potential chemical effect on flux measurement in Section 2.2.2 and the calculation method of $\tau_{chem}$ in supplemental material.

Lines 132-134: "Total residence time of air samples from the tip of the inlet to the point of $O_3$ detection was about 10 s, which was much shorter that the lifetime of $O_3$ reacting with NO (Supplementary methods), suggesting the chemical reaction in the two channels could be neglected."

**Supplementary Methods: Evaluation of the chemical reaction impacts on $O_3$ flux measurement.**

The comparison between the total residence time of the REA system and the chemical reaction time was used to evaluate to the influence of the chemical reaction on $O_3$ flux measurement. In the two light-shielded channels, $O_3$ mainly reacted with NO, which could occur in minutes in agricultural fields (Zhang et al., 2024). In our study, since there was no simultaneous observation of NO concentrations in the two channels, we estimated the lifetime ($\tau_{chem}$) of $O_3$ reacting with NO using ambient NO concentrations, which should be a lower limit to the actual $\tau_{chem}$ in the two channels.

$$\tau_{chem} = \frac{1}{C_{NO} \times k_r}, \qquad \text{Eq. (S1)}$$

$$k_r = 0.444 \times \exp\left(\frac{1360}{(T_{Air}+273.15)}\right), \qquad \text{Eq. (S2)}$$

where $C_{NO}$ is NO concentration in ppb, $k_r$ is the reaction rate constant in ppb$^{-1}$ s$^{-1}$ (Walton et al., 1997) and $T_{Air}$ is the air temperature in °C

From 18 March to 2 June, NO concentration ranged from 0.1 ppb to 23.3 ppb with an average of 0.8 ± 1.8 ppb. The estimated $\tau_{chem}$ of $O_3$ in the atmosphere ranged from 141 s to 31559 s, with an average of 8893 s, which was remarkably higher than the total residence time (~ 10 s) from tip of the inlet to the point of $O_3$ detection in the REA system. This indicated that the chemical $O_3$ consumption in the two channels was minor and had marginal effects on the flux measurements.

(4) The zero air is supplied by an external air compressor and a zero air supply (Model 111, Thermo Fisher Scientific Inc., USA). Model 111 can supply pollutant-free air from the ambient air to allowing for proper zeroing. The components to be removed include $SO_2$, $O_3$, NO, $NO_2$, CO and hydrocarbons. Since water vapor is not an air pollutant, the Model 111 does not have a drying system, but the air compressor will reduce the dew point as a result of air compression. We added the detailed description of zero air in Lines 124-127:

"The zero air was generated by an external air compressor (M104, Gast Manufacturing Inc., USA) and a zero-air generator (Model 111, Thermo Fisher Scientific Inc., USA), which removes $O_3$, NO, $NO_2$, CO and hydrocarbons from ambient air, but not water vapor."

Reference:

Hicks, B. and McMillen, R.: A Simulation of the Eddy Accumulation Method for Measuring Pollutant Fluxes, Journal of Applied Meteorology, 23, 637-643, 10.1175/1520-0450(1984)023<0637:ASOTEA>2.0.CO;2, 1984.

Zhang, C., Wang, J., Zhang, Y., Xu, W., Zhang, G., Miao, G., Zhou, J., Yu, H., Zhao, W., Lin, W., Kang, L., Cai, X., Zhang, H., and Ye, C.: Improving model representation of rapid ozone deposition over soil in the central Tibetan Plateau, Environmental Science: Atmospheres, 4, 252-264, 10.1039/D3EA00153A, 2024.

On the comparison of the two channels by simultaneous measurements and the calibrations: Since two separate analysers are used and concentration measurements are subtracted from each other, measurement uncertainties from each analyser have to be considered and estimated. The Thermo Fisher 49i analysers are expected to be very stable in their sensitivity, so the monthly multi-point calibration is commendable. Nevertheless, the stability of the two instruments against each other is of critical importance for the quality of the calculated fluxes, therefore I would suggest adding this information on sensitivities and offsets from the multipoint calibrations in the supplementary material. Information on uncertainties in sensitivity should be added so that Figure S2 (and the slope of 1.02 and offset of 1.05) can be evaluated as to whether they are within the measurement uncertainty (2 sigma) or not. Does Figure S2 show 1-minute or 30-minute averaged data? Over what time frame and at what frequency where these comparison measurements made?

**Response**: Thanks for your suggestions. We added the calibration results in Figure S4, and the determination coefficients ($R^2$) of the calibration curves were 1.000, indicating the stability of the two channels and $O_3$ instruments.

Lines 171-174: "Figure S4 presents the results of multipoint calibration, with determination coefficients ($R^2$) reaching 1.000, indicating high stability of the two channels and $O_3$ instruments. $O_3$ concentration in the two channels were adjusted based on the results of parallel experiments and standard calibrations."

[Figure]

**Figure S4. The results of multipoint calibration in the updraft and downdraft reservoirs.**

The parallel experiment of the two channels of the REA system was conducted through the simultaneous measurements of ambient $O_3$ by sampling air directly from the system inlet during 10-11 May. Figure S3 presents the comparison of 1-minute $O_3$ concentration measured by the two channels. A total of 1168 pairs of $O_3$ data were obtained in the parallel experiment, which exhibited high consistency with the slope of 1.02 (p < 0.01), suggesting that the systematic difference in measured $O_3$ was minimal between updraft and downdraft reservoirs and that its impact on flux measurements could be ignored. In addition, the consistency of the REA system was checked monthly based on the results of multipoint calibrations. The standard $O_3$ concentrations generated by an $O_3$ calibrator (TE 49C PS, Thermo Fisher Scientific Inc., USA) were introduced into the system from the zero air inlet and simultaneously measured by the two $O_3$ analyzers.

We added the sensitivity results of the parallel experiment in Figure S3, and revised the discussion as follows:

Lines 164-169: "To identify potential flux errors induced by any differences between updraft and downdraft channels in the REA-$O_3$ system (including valves, sample tubes and $O_3$ analyzers), the parallelism of the two sampling channels was checked through simultaneous direct sampling of ambient $O_3$ from the REA system inlet during 10-11 May. As shown in Figure S2, $O_3$ data points obtained from the two channels all aligned close to the 1:1 line (slope = 1.02, p < 0.01), suggesting that the difference in measured $O_3$ was minimal between updraft and downdraft reservoirs and that its impact on flux measurement could be ignored."

[Figure]

**Figure S3. The comparison of O₃ concentration in the updraft and downdraft reservoirs during 10-11 May.**

2.3. Field measurements and ancillary data

NO$_x$ measurements: add more information on the location/distance/height of the NO$_x$ inlet with respect to the ozone measurements. No plot in the manuscript shows a direct comparison (timeseries) of both ozone and NO$_x$ (NO and NO₂ speciated) for the campaign time period, however this could orient the reader as to the nature of pollution regime at the site. Consider adding such information in the supplementary material.

**Response**: Thanks for the suggestions. The NO$_x$ concentrations were measured on the northern edge of the cropland, which is located 200 m north of the O₃ measurement. The inlet of NO$_x$ analyzer is installed 1.8 m above the roof of the container (~ 4.5 m above ground level). During the NO$_x$ observation (from 18 March to 2 June), both NO and NO₂ concentration revealed large fluctuations, implying high variability in air pollution at the GC site.

We added a description of the location and height of the NO$_x$ measurement and a brief summary of observed NO$_x$ concentrations in Section 2.2, and supplied the timeseries of NO and NO₂ concentration in Figure S6.

Lines 197-204: "NO$_x$ (NO/NO₂/NO$_x$) concentrations were monitored from 18 March to 2 June by a NO-NO₂-NO$_x$ Trace Level Analyzer (Model 42C-TL, Thermo Fisher Scientific Inc., USA). The analyzer is installed in an air-conditioned container on the northern edge of the cropland, which is located 200 m north of the eddy covariance tower. The air inlet is 1.8 m above the roof of the container and ~ 4.5 m above ground level, with an estimated residence time of less than 3 s. Multipoint calibrations of NO$_x$ were made using a NO standard gas obtained from National Institute of Metrology, Beijing, China. A total set of 3631 30-min averaged NO$_x$ data were obtained, as shown in Figure S6. During the measurement period, NO concentration ranged from 0.1 ppb to 23.3 ppb with an average of 0.8 ± 1.8 ppb, while NO₂ ranged from 0.5 ppb to 6.6 ppb with an overall average of 2.3 ± 0.9 ppb."

[Figure]

**Figure S6. Timeseries of NO (a) and NO$_2$ (b) concentration from 18 March to 2 June.**

3 Results and discussion

3.1 Meteorological conditions

Is information available from the farmers when irrigation events occurred and also how they compare to the recorded precipitation events (volume water applied)? Figure 2 is incomplete in that respect and the reader can only speculate as to how much irrigation water was applied to achieve the steep increases in soil VWC.

**Response**: During the main growth stage of wheat, cropland experienced two irrigation events that occurred during 19-21 April and 18-22 May, respectively. The irrigation method was sprinkler irrigation. The total area of the cropland is 50 acres. The irrigation is divided into 3 times of watering, and the duration time of each watering is 24 h. The outlet flow of the irrigation pump is 50 m$^3$ h$^{-1}$, and a total of 20 sprinklers are connected to the pump and watering at the same time. Therefore, the total volume water of the irrigation is 3600 m$^3$ for 33333 m$^2$ cropland, which is equivalent to a 3-day continuous precipitation event with a daily average rainfall of 36 mm. We annotated the irrigation events (black arrows) in Figure 2c, which corresponded to the steep increases in soil VWC on 20 April and 19 May, respectively. We added a detailed description of the irrigations in Section 3.1.

Lines 228-231: "During the wheat growth stage, the cropland experienced two irrigation events, occurring on 19-21 April and 18-22 May, respectively (Figure 2c). The irrigation system mainly consists of an irrigation pump with an outlet flow of 50 m$^3$ h$^{-1}$ and a total of 20 sprinklers. The total irrigation water volume was approximately 3600 m$^3$ for 33333 m$^2$ cropland, and the duration was 3 days."

[Figure]

**Figure 2: Daily meteorological and soil conditions from 12 February to 18 June 2023: (a) T$_{Air}$ and RH, (b) u$_*$ and VPD, (c) irrigation (black arrows), precipitation, T$_{Soil}$ and soil VWC, (d) G and PAR.**

3.2 O$_3$ flux, deposition and concentration

Figure 3. see comment above on the additional information of NO$_x$. Relevance here, ozone concentrations show a considerable range from 0 to 280 ug/m³.

**Response**: The O$_3$ flux and concentration could be synchronously measured by the new REA-O$_3$ flux system, which was mounted at the height of 4.5 m in an eddy covariance tower in the middle of cropland. The REA-O$_3$ flux system directly measured the averaged O$_3$ concentration in the updraft and downdraft reservoirs. The actual O$_3$ concentration in the updraft and downdraft were calculated using the sampling time, sample flow and observed O$_3$ concentrations, according to Eq. (4). The ambient O$_3$ concentration was calculated from the average of O$_3$ concentration in the updraft and downdraft ($C_{O_3} = \frac{\overline{c^+} + \overline{c^=}}{2}$).

The following was added to describe the locations of O$_3$ and NO$_x$ measurement, and the calculation method of O$_3$ concentration in the REA-O$_3$ flux system:

Line 111-113: "A 3-D sonic anemometer (CSAT3, Campbell Scientific Inc., USA) was used for measuring the three wind components (*u*, *v*, *w*) at 10 Hz, which was amounted at the height of 4.5 m on an eddy covariance tower, located in the middle of cropland."

Lines 198-200: "The analyzer is installed in an air-conditioned container on the northern edge of the cropland, which is located 200 m north of the eddy covariance tower. The air inlet is 1.8 m above the roof of the container and ~ 4.5 m above ground level, with an estimated residence time of less than 3 s."

Lines 141-143: "The O$_3$ concentration was calculated by averaging O$_3$ concentrations under updraft and downdraft conditions using Eq. (5)."

$$C_{O_3} = \frac{\overline{c^+} + \overline{c^-}}{2}, \qquad\qquad\qquad\qquad\qquad\qquad\qquad\qquad\qquad\qquad\qquad \text{Eq. (5)}"$$

Please provide an interpretation of the observed ozone emission flux on 15 March 2023.

**Response**: During the observation period, positive $O_3$ fluxes (emission fluxes) were observed in several cases, mainly occurring during nighttime. The largest positive $O_3$ flux (0.14 μg m$^{-2}$ s$^{-1}$) was detected during midnight on 15 March 2023. At night, the earth surface undergoes rapid radiative cooling and the air near ground surface cools down faster, easily causing temperature inversions. Both wind and friction velocity sharply decreased, and atmosphere turbulence weakened, inhibiting the vertical movement of air parcels during nighttime. Therefore, $O_3$ deposition reached the minimum level at night, as shown by the diurnal changes of $O_3$ deposition. Meanwhile, $O_3$ concentration also dropped to the lowest level due to the absence of photochemical production and strong NO titration at night. Nocturnal $O_3$ levels are mainly dependent on regional transport, daytime photochemical residuals and consumptions (dry deposition and NO titration) (Zhu et al., 2020). Under certain circumstances (such as horizontal transport), the levels of NO might be higher in the upper layer, which could lead to a more intense titration consumption of $O_3$ at higher levels than within the wheat canopy, temporarily altering the vertical distribution and deposition of $O_3$. Besides, the small positive $O_3$ fluxes might be attributed to the uncertainty of the REA system, when $O_3$ concentration differences in the updrafts and downdrafts closed to zero in rare cases. We revised "the largest $O_3$ deposition and emission fluxes" as "the largest negative and positive $O_3$ flux" in Lines 241-243:

"The largest negative (-3.20 μg m$^{-2}$ s$^{-1}$) and positive (0.14 μg m$^{-2}$ s$^{-1}$) fluxes were measured during noontime on 29 April and during midnight on 15 March, respectively."

Reference:

Zhu, X., Ma, Z., Li, Z., Wu, J., Guo, H., Yin, X., Ma, X., and Qiao, L.: Impacts of meteorological conditions on nocturnal surface ozone enhancement during the summertime in Beijing, Atmos. Environ., 225, 117368, https://doi.org/10.1016/j.atmosenv.2020.117368, 2020.

Line 220. Consider the word "uncertainties" of the reported mean values. Are uncertainties or observed variability around the mean reported in the studies mentioned?

**Response**: We apologize for the ambiguity in our expression that may have caused a misunderstanding. In the original manuscript, the term "uncertainties" referred to the comparison uncertainties, as the large difference of observation time, methods, land covers, regions of the reported $O_3$ deposition velocities. We deleted the sentence "considering the large uncertainties of reported mean values".

In the mentioned studies, only Wu et al. (2015) compared observed $O_3$ deposition fluxes over a forest canopy using three flux-gradient methods (aerodynamic gradient method (AGM), the modified Bowen ratio method (MBR) and the modified micrometeorological gradient method (MGM)), and investigated the uncertainties of the MGM, which were mainly from the input parameters (the wind speed attenuation, canopy displacement height and leaf area density vertical profiles).

Reference:

Wu, Z. Y., Zhang, L., Wang, X. M., and Munger, J. W.: A modified micrometeorological gradient method for estimating O$_3$ dry depositions over a forest canopy, Atmos. Chem. Phys., 15, 7487-7496, 10.5194/acp-15-7487-2015, 2015.

Figure 4. Considering that the time period February to June 2023 shows changes in time in ozone concentration and also meteorological and soil variables (Fig. 2), is the inference robust, i.e. do the median values show the same pattern? Since H$_2$O and CO$_2$ flux measurements were also made, the statement that stomatal opening is driving the ozone deposition could be supported if the CO$_2$ and H$_2$O flux variation is consistent with Figure 4. The partitioning into stomatal and non-stomatal ozone deposition could be attempted (see e.g. Fares, S., Matteucci, G., Scarascia Mugnozza, G., Morani, A., Calfapietra, C., Salvatori, E., Fusaro, L., Manes, F., Loreto, F., 2013b. Testing of models of stomatal ozone fluxes with field measurements in a mixed Mediterranean forest. Atmos. Environ. 67, 242–251.)

**Response**: Yes, the median values of O$_3$ deposition showed a similar diurnal pattern with the averaged values, as shown in Figure R1. O$_3$ deposition rapidly increased in the morning, and both O$_3$ deposition flux and velocity reached their peaks (-0.51 μg m$^{-2}$ s$^{-1}$ and 0.49 cm s$^{-1}$) by 13:00. Afterwards, O$_3$ deposition quickly decreased from 14:00 to 19:00, and kept at low levels at night with an averaged flux and velocity of -0.04 ± 0.04 μg m$^{-2}$ s$^{-1}$ and 0.12 ± 0.02 cm s$^{-1}$, respectively.

[Figure]

**Figure R1. Diurnal variations of O$_3$ deposition flux, V$_d$ and concentration during the wheat growing season, with error bars representing median values ± quantiles.**

Firstly, we added diurnal changes of cropland H$_2$O, CO$_2$ fluxes, PAR and u$_*$ during the wheat growth season in Figure 5. H$_2$O and CO$_2$ fluxes exhibited similar diurnal patterns as that of O$_3$ deposition, implying the dominant role of stomatal uptake in O$_3$ deposition over the wheat fields. We integrated the diurnal patterns of O$_3$ deposition with the observed cropland fluxes and meteorological factors, and revised the discussion as follows:

"With solar radiation and atmospheric turbulence increasing after sunrise, plant stomatal conductance increased along with H$_2$O, CO$_2$ fluxes over the cropland, reaching peaks at noon (Figure 5). O$_3$ deposition rapidly rose during the morning (06:00-10:00). Deposition flux and velocity both reached their peaks (-0.62

μg m$^{-2}$ s$^{-1}$ and 0.54 cm s$^{-1}$) by 13:00, when stomatal conductance and gas-leaf exchange also reached their diurnal maximums (Rannik et al., 2012; Otu-Larbi et al., 2021). $O_3$ deposition quickly decreased from 14:00 to 18:00 despite of high levels of $O_3$ (Figure 4). At night, atmospheric turbulence weakened and leaf stomata closed, resulting in reduced $H_2O$ and $CO_2$ fluxes keeping steady throughout the night (Figure 5). Nighttime $O_3$ deposition also remained at relatively low levels and exhibited weak changes, with an averaged flux and $V_d$ of -0.09 ± 0.04 μg m$^{-2}$ s$^{-1}$ and 0.17 ± 0.02 cm s$^{-1}$, respectively. Therefore, diel variations in $O_3$ deposition over the wheat fields were mainly driven by the stomatal opening and closing, with $O_3$ deposition velocity being decisive of deposition flux diurnal variations."

[Figure]

**Figure 5. Diurnal variations of (a) $H_2O$ flux ($F_{H_2O}$), $CO_2$ flux ($F_{CO_2}$), (b) $u_*$ and PAR during the wheat growing season, with error bars representing average ± standard deviation/$\sqrt{n}$.**

The variation of $CO_2$ flux in the agricultural ecosystem from February to June 2023 was also added in Figure 6. The temporal changes of $V_d$ were synchronous with the evolutions of cropland $CO_2$ flux, suggesting the determining effect of crop growth on $O_3$ deposition and the predominant contribution of stomatal uptake over wheat fields during its growth season. We adjusted the discussion in Section 3.3 as follows:

Lines 273-282: "To investigate the influences of wheat growth on $O_3$ deposition, the characteristics of $O_3$ deposition were further examined in connection to the different growth stages. During the O-W stage, wheat was in dormancy and leaves had not begun to turn green (LAI < 0.5, Figure 6b), with $CO_2$ flux in the agricultural ecosystem closed to zero (Figure 6c). $V_d$ in the O-W stage barely changed, exhibiting a low average value of 0.20 ± 0.28 cm s$^{-1}$ and a median of 0.12 cm s$^{-1}$ (Table 1). Wheat grew vigorously in the G-F stage, with LAI and $CO_2$ assimilation flux exhibiting rapid increases until the early and late flowering stage, respectively, after which both of them gradually decreased (Figure 6b-c). $O_3$ deposition varied nearly in synchronization with LAI and wheat growth, with $V_d$ reaching a peak when cropland $CO_2$ assimilation was the highest during the G-F stage (Figure 6a), reaching highest daytime and nighttime averages of 0.46 ± 0.41 cm s$^{-1}$ and 0.24 ± 0.28 cm s$^{-1}$, respectively (Table 1). Afterwards, with the maturing of wheat and the aging of leaves in the R-H stage, $V_d$ quickly dropped back to a low average level of 0.20 ± 0.25 cm s$^{-1}$, similar to that observed in the O-W stage."

[Figure]

**Figure 6.** (a) $O_3$ $V_d$, (b) LAI and FPAR, (c) $CO_2$ flux ($F_{CO_2}$) in different wheat growing stages. The circles and error bars in (a) denote the weekly medians and quantiles of $V_d$, respectively. O-W, G-F and R-H represent Over-Wintering, Green-Flowering and Ripening-Harvest stages.

We are thankful for the suggestion to attempt partitioning observed $O_3$ deposition into stomatal and non-stomatal deposition. We acknowledge the necessity and importance of such partitioning, which is the focus of our upcoming study that is already in preparation. In this study, we evaluated the feasibility of the new REA-$O_3$ flux system, and presented $O_3$ deposition and flux measurements throughout a whole wheat growth season to confirm that measurements are robust and credible. We focused on the variation characteristics of $O_3$ deposition over the wheat canopy and their key environmental drivers during the different wheat growth stages. These contents have already summed up to a fairly long manuscript, which is why we decided to make the partitioning of $O_3$ deposition the main body of our next work. We have already calculated the different $O_3$ deposition resistances following deposition parameterization methods in previous literatures. However, no definitive conclusions have been reached yet on the relative contributions of stomatal and non-stomatal $O_3$ deposition. To acknowledge the importance and necessity of this partitioning, we added the following to the conclusions and implications section:

Lines 427-447: "However, the relative contributions of stomatal and non-stomatal $O_3$ deposition pathways need to be further quantified in our future investigations, which is of crucial importance for studies on $O_3$ removal and vegetation health impacts. While the influence of crops on $O_3$ deposition through stomatal uptake or surface removal has been extensively investigated in previous studies (Ainsworth, 2017; Aunan et al., 2000; Bender and Weigel, 2011; Biswas et al., 2008; Epa, 2013; Felzer et al., 2005; Harmens et al., 2018; Piikki et al., 2008), those of

O$_3$ deposition on crop growth and yield under currently rising O$_3$ levels in the NCP remains an unsolved issue. Many researches have assessed the crop yield loss induced by O$_3$ pollution based on exposure-response functions (Feng et al., 2019; Hu et al., 2020; Feng et al., 2020). However, the actual exposure is more related to direct deposition flux measurements rather than concentration-based indicators (Zhu et al., 2015). Therefore, agricultural impacts of O$_3$ should be more accurately quantified using stomatal O$_3$ deposition fluxes that might be obtained from current total O$_3$ deposition flux measurements using partitioning methods such as those in Fares et al. (2013) in our following studies."

3.4 O$_3$ deposition relation to environmental factors

Results from the stepwise MLR model are described for each growth stage (page 12), but elaboration on possible reasons (physical, physiological, chemical) for the factors and differences between the stages could be helpful to understand processes driving such relationships.

**Response**: Thanks for your suggestions, we further investigated the underlying reasons of the specific key factors for O$_3$ deposition for different wheat growth stages. During the Over-Wintering stage, LAI was lower than 0.5, thus wheat growth played a marginal role in O$_3$ deposition. O$_3$ deposition was more sensitive to u$_*$, soil VWC and PAR. That is because solar radiation (PAR) and wind (u$_*$) supplied energy for atmospheric turbulence, which effectively transported O$_3$ to soil surface, while soil moisture (soil VWC) largely affected the diffusion and absorption of O$_3$ in soil. In the Green-Flowering stage, wheat grew vigorously and leaf stomatal uptake had a significant effect on total O$_3$ deposition, thus LAI was determined to be the most important factor for O$_3$ deposition. In the Ripening-Harvest stage, canopy surface was mainly covered by mature wheat with gradually decreasing LAI. At this stage O$_3$ stomatal deposition decreased and was mainly dependent on turbulent transport, which was mostly affected by T$_{Air}$ and u$_*$ under sufficient summertime solar radiation. During nighttime, O$_3$ deposition mainly commenced through non-stomatal pathways, such as cuticular and soil deposition (Xu et al., 2018), which were affected by surface conditions (T$_{Air}$, T$_{Soil}$ and soil VWC) and turbulence strength (u$_*$). We added the possible reasons for the results of stepwise MLR models to Lines 302-313:

"Distinct key environmental factors for O$_3$ deposition were identified for different wheat growth stages, while LAI was only among the most important factors during the G-F stage (Table 2), confirming the significant effect of crops on O$_3$ deposition during its most vigorous growing stage. During the O-W stage, wheat played a minor role in O$_3$ deposition as LAI < 0.5. Solar radiation (PAR) and wind (u$_*$) supplied energy for atmospheric turbulence, which transported O$_3$ to soil surface, while soil moisture (soil VWC) largely affected the diffusion and absorption of O$_3$ in soil (Stella et al., 2011). Therefore, O$_3$ deposition was more sensitive to u$_*$, soil VWC and PAR during the O-W stage. In the R-H stage, the land surface was covered by ripe wheat, which reduced LAI and stomatal conductance. O$_3$ deposition was therefore mainly dependent on turbulent transport, which was more affected by T$_{Air}$ and u$_*$ under sufficient solar radiation (Fig. 2d). During nighttime, O$_3$ deposition mainly commenced through non-stomatal pathways such as cuticular and soil deposition (Xu et al., 2018), that are affected by turbulence strength and surface condition. The most significant influencing factors for O$_3$ deposition were T$_{Air}$, u$_*$, T$_{Soil}$ and soil VWC for the whole observation period (Table 2)."

Line 290-295. Can any further information from the literature be provided on stomatal response in wheat and effects related to humidity (not the already quoted Zhu et al., 2014; Zhu et al., 2015 in line 288)?

**Response**: Yes, the response difference of leaf stomata to humidity might be largely attributed to wheat cultivars. Kudoyarova et al. (2007) compared the stomatal responses of two wheat cultivars to atmospheric humidity based on labotorary and field experiments. The results showed that the stomatal conductance of wheat cultivar, which was relatively drought-tolerant, declined rapidly with reduced humidity, whereas in high-productivity cultivar, leaf stomata could open wider in response to reduced humidity, which were affected by temperature and soil water contents. The response differences were related to the pattern of hormone (ABA) partition. Accumulation of ABA in the vicinity of stomata could result in their closure, while ABA in the roots should result in the increase of hydraulic conductance (Kudoyarova et al., 2011; Morillon and Chrispeels, 2001). In addition, the transpiration of wheat generally increased with VPD, affecting the leaf gas exchange, with response patterns being strongly dependent on wheat genotypes (Ranawana et al., 2021). On the other hand, aerodynamically rough surfaces under relatively dry conditions (high VPD) could induce further turbulent transfer, transporting gases more effectively to the leaf surface thereby promoting $O_3$ deposition (Liao et al., 2022). Therefore, high-yield wheat land generally exhibited increased $O_3$ deposition velocity with the decrease of RH. We modified the discussion of $O_3$ deposition response to humidity changes as in the following:

Lines 326-342: "Ambient RH influences the relative contributions of stomatal and non-stomatal processes to $O_3$ deposition. A clear negative variation of $V_d$ with increasing RH was observed at GC both for daytime and nighttime (Figure 7a and 9b). Due to the negative correlation of VPD to RH (Eq.7), high VPD was conducive to high nighttime $V_d$ (Figure 9c). During daytime, this negative relationship was detected for all growth stages, while during nighttime the decrease with RH was most evident during the G-F stage. At RH above 60-70%, leaf surfaces are frequently covered by a thin liquid film or by dew water, which would inhibit stomatal dry deposition, but enhance aqueous reactions of $O_3$, leading to enhanced relative contribution of non-stomatal deposition, however, with high variability (Coyle et al., 2009; Lamaud et al., 2009). Under RH below 60%, stomatal conductance contributes more to $O_3$ deposition, which might also be negatively dependent on RH. For instance, similar negative correlations of $V_d$ and RH were observed over wheat and maize canopy in the NCP (Zhu et al., 2014; Zhu et al., 2015), while $O_3$ deposition into a boreal forest revealed strong positive correlation with RH (Rannik et al., 2012), which was attributed to differences in plant varieties and growth environment. The response of stomata to the changes in humidity is largely dependent on plant cultivars and plant water stress (Camacho et al., 1974; Rawson et al., 1977; Fanourakis et al., 2020). For example, the stomata of drought-tolerant wheat cultivar closed rapidly with reduced humidity, whereas high-yield cultivar stomatal conductance increased against decreasing humidity (Kudoyarova et al., 2007). The growth environment affects the aerodynamically roughness of the earth surfaces. Increased roughness could induce stronger turbulent transfer under low humidity condition, transporting gases more effectively to the leaf surfaces, thereby promoting $O_3$ deposition (Liao et al., 2022)."

Reference:

Kudoyarova, G., Veselova, S., Hartung, W., Farhutdinov, R., Veselov, D., and Sharipova, G.: Involvement of root ABA and hydraulic conductivity in the control of water relations in wheat plants exposed to increased evaporative demand, Planta, 233, 87-94, 10.1007/s00425-010-1286-7, 2011.

Kudoyarova, G. R., Veselov, D. S., Faizov, R. G., Veselova, S. V., Ivanov, E. A., and Farkhutdinov, R. G.: Stomata response to changes in temperature and humidity in wheat cultivars grown under contrasting climatic conditions, Russian Journal of Plant Physiology, 54, 46-49, 10.1134/S1021443707010074, 2007.

Liao, Q., Ding, R., Du, T., Kang, S., Tong, L., and Li, S.: Stomatal conductance drives variations of yield and water use of maize under water and nitrogen stress, Agricultural Water Management, 268, 107651, https://doi.org/10.1016/j.agwat.2022.107651, 2022.

Morillon, R. and Chrispeels, M. J.: The role of ABA and the transpiration stream in the regulation of the osmotic water permeability of leaf cells, Proc Natl Acad Sci U S A, 98, 14138-14143, 10.1073/pnas.231471998, 2001.

Ranawana, S. R. W. M. C. J. K., Siddique, K. H. M., Palta, J. A., Stefanova, K., and Bramley, H.: Stomata coordinate with plant hydraulics to regulate transpiration response to vapour pressure deficit in wheat, Functional Plant Biology, 48, 839-850, https://doi.org/10.1071/FP20392, 2021.

Line 355-370. As suggested above, it would enhance the paper if stomatal conductance (stomatal ozone flux) is estimated, which would strengthen the evidence for statements made on relationships and dependencies. Partitioning/estimating also non-stomatal ozone deposition would allow to address and provide possible support on NO soil emission, as conjectured here.

**Response**: We totally agree with your comments. The partitioning into stomatal and non-stomatal $O_3$ deposition could enhance the understandings for the $O_3$ deposition response to the environmental factors, and provide supports on the possible biotic and chemical causes of $O_3$ deposition changes after a sudden increase in soil moisture, as discussed in Section 3.4. However, we want to focus on the variability and related factors of $O_3$ deposition flux and velocity over wheat fields in the North China Plain in this study, and the partitioning into stomatal and non-stomatal $O_3$ deposition is expected to be the focus and main body of our next work.

4 Conclusions and implications

Make changes to overall conclusions in accordance with revisions in the main text.

**Response**: Thanks for your suggestion. We revised the conclusion as follows:

"In this study, we developed a relaxed eddy accumulation (REA) $O_3$ flux measurement system, verified its reliability, and conducted measurements of $O_3$ deposition using this newly developed REA system over the wheat canopy at a polluted agricultural site (GC) in the NCP during the main wheat growth season. Ancillary data related to $O_3$ deposition and made an integrated analysis on the environmental influencing factors. The observed $O_3$ deposition flux and velocity over the wheat fields at GC reached averages of $-0.25 \pm 0.39$ μg m$^{-2}$ s$^{-1}$ and $0.29 \pm 0.33$ cm s$^{-1}$, respectively. The diurnal cycle of $V_d$ was controlled by the crop stomatal opening and turbulent transport during the day. $V_d$ was obviously higher during daytime ($0.40 \pm 0.38$ cm s$^{-1}$) than nighttime ($0.17 \pm 0.26$ cm s$^{-1}$). $V_d$ played a decisive role in the diel pattern of $O_3$ deposition flux, while $O_3$ concentrations determined the flux variability on longer timescales. The temporal changes of $V_d$ were synchronous with the evolutions of LAI, wheat growth and cropland $CO_2$ flux, suggesting the determining and enhancement effect of crop growth on $O_3$ deposition and the predominant contribution of stomatal uptake over wheat fields during its growth season. However, the relative contributions of stomatal and non-stomatal $O_3$

deposition pathways need to be further quantified in our future investigations, which is of crucial importance for studies on $O_3$ removal and vegetation health impacts. While the influence of crops on $O_3$ deposition through stomatal uptake or surface removal has been extensively investigated in previous studies (Ainsworth, 2017; Aunan et al., 2000; Bender and Weigel, 2011; Biswas et al., 2008; Epa, 2013; Felzer et al., 2005; Harmens et al., 2018; Piikki et al., 2008), those of $O_3$ deposition on crop growth and yield under currently rising $O_3$ levels in the NCP remains an unsolved issue. Many researches have assessed the crop yield loss induced by $O_3$ pollution based on exposure-response functions (Feng et al., 2019; Hu et al., 2020; Feng et al., 2020). However, the actual exposure is more related to direct deposition flux measurements rather than concentration-based indicators (Zhu et al., 2015). Therefore, agricultural impacts of $O_3$ should be more accurately quantified using stomatal $O_3$ deposition fluxes that might be obtained from current total $O_3$ deposition flux measurements using partitioning methods such as those in Fares et al. (2013) in our following studies.

During the wheat growth season, RH, $u_*$, soil VWC and LAI were identified as the most significant factors in explaining the changes of $O_3$ deposition during daytime through stomatal and non-stomatal pathways, while $u_*$ and soil VWC were more important for nocturnal $O_3$ deposition, which mainly commenced through non-stomatal deposition. $V_d$ significantly increased with the decrease of RH and the increases of $u_*$, PAR and soil VWC, especially under higher LAI. Rapid increases of soil VWC after strong precipitation or irrigation events extended stomatal opening to nighttime hours, leading to increased stomatal conductance, enhanced transpiration and photosynthesis of wheat, which remarkably strengthened $O_3$ stomatal uptake during daytime and nighttime. Stomatal opening and transpiration are typically assumed to occur specifically during daytime. However, an increasing number of studies have shown the non-negligible effects of unclosed nocturnal stomata and transpiration for a wide range of plant species (Kukal and Irmak, 2022; Schoppach et al., 2020; Tamang et al., 2019; Ramírez et al., 2018; Hoshika et al., 2018). Therefore, how nocturnal plant activities would interact with the significantly increasing nighttime $O_3$ levels in China during recent years (Agathokleous et al., 2023; He et al., 2022) is also worth deeper investigations. Aside from influencing stomata opening, drastic changes in soil humidity also strengthened NO soil emissions, facilitating NO titration of $O_3$ within the canopy and enhancing non-stomatal $O_3$ removals at night. Therefore, drastically increasing soil moisture simultaneously led to strong increments in $V_d$. Under current climate warming trends, extreme weather events (such as extreme precipitation and drought) have gained in frequency in agricultural areas (Yuan et al., 2016), and its effect on agriculture, $NO_x$ emissions, $O_3$ formation as well as $O_3$ deposition require future attentions.

During the entire wheat growth season, $O_3$ deposition velocity exhibited large fluctuations under changing environmental conditions, with distinct factors determining $V_d$ variability during different wheat growth stages. These key influencing factors and their effects on $O_3$ deposition would also vary with canopy types and ground surface conditions. Aside from environmental conditions, agricultural activities also significantly affect $O_3$ deposition (Mészáros et al., 2009). Therefore, the actual $O_3$ deposition process bears large uncertainties and varies greatly in space and time. More $O_3$ deposition observations over different types of land surfaces and vegetations are urgently needed to facilitate the exploration on $O_3$ dry deposition mechanisms and to optimize current model parameterizations whose results largely deviate from observed $O_3$ dry deposition fluxes in crop growth seasons (Clifton et al., 2020; Hardacre et al., 2015)."

Line 414. Assessment of deposition parameterization schemes have so far not been content of this manuscript, therefore the statement on possible large errors in simulation results does not follow from the analysis presented in this study. Some of the statements in this and final paragraph are rather vague or expressed in too

general a way. Consider revising this and the final paragraph to more directly follow from the results shown here.

**Response**: Thanks for your suggestion, we have adjusted the implications so that they are more relevant to the content of the study. The conclusions section was revised as:

"In this study, we developed a relaxed eddy accumulation (REA) $O_3$ flux measurement system, verified its reliability, and conducted measurements of $O_3$ deposition using this newly developed REA system over the wheat canopy at a polluted agricultural site (GC) in the NCP during the main wheat growth season. Ancillary data related to $O_3$ deposition and made an integrated analysis on the environmental influencing factors. The observed $O_3$ deposition flux and velocity over the wheat fields at GC reached averages of -0.25 ± 0.39 μg m$^{-2}$ s$^{-1}$ and 0.29 ± 0.33 cm s$^{-1}$, respectively. The diurnal cycle of $V_d$ was controlled by the crop stomatal opening and turbulent transport during the day. $V_d$ was obviously higher during daytime (0.40 ± 0.38 cm s$^{-1}$) than nighttime (0.17 ± 0.26 cm s$^{-1}$). $V_d$ played a decisive role in the diel pattern of $O_3$ deposition flux, while $O_3$ concentrations determined the flux variability on longer timescales. The temporal changes of $V_d$ were synchronous with the evolutions of LAI, wheat growth and cropland $CO_2$ flux, suggesting the determining and enhancement effect of crop growth on $O_3$ deposition and the predominant contribution of stomatal uptake over wheat fields during its growth season. However, the relative contributions of stomatal and non-stomatal $O_3$ deposition pathways need to be further quantified in our future investigations, which is of crucial importance for studies on $O_3$ removal and vegetation health impacts. While the influence of crops on $O_3$ deposition through stomatal uptake or surface removal has been extensively investigated in previous studies (Ainsworth, 2017; Aunan et al., 2000; Bender and Weigel, 2011; Biswas et al., 2008; Epa, 2013; Felzer et al., 2005; Harmens et al., 2018; Piikki et al., 2008), those of $O_3$ deposition on crop growth and yield under currently rising $O_3$ levels in the NCP remains an unsolved issue. Many researches have assessed the crop yield loss induced by $O_3$ pollution based on exposure-response functions (Feng et al., 2019; Hu et al., 2020; Feng et al., 2020). However, the actual exposure is more related to direct deposition flux measurements rather than concentration-based indicators (Zhu et al., 2015). Therefore, agricultural impacts of $O_3$ should be more accurately quantified using stomatal $O_3$ deposition fluxes that might be obtained from current total $O_3$ deposition flux measurements using partitioning methods such as those in Fares et al. (2013) in our following studies.

During the wheat growth season, RH, $u_*$, soil VWC and LAI were identified as the most significant factors in explaining the changes of $O_3$ deposition during daytime through stomatal and non-stomatal pathways, while $u_*$ and soil VWC were more important for nocturnal $O_3$ deposition, which mainly commenced through non-stomatal deposition. $V_d$ significantly increased with the decrease of RH and the increases of $u_*$, PAR and soil VWC, especially under higher LAI. Rapid increases of soil VWC after strong precipitation or irrigation events extended stomatal opening to nighttime hours, leading to increased stomatal conductance, enhanced transpiration and photosynthesis of wheat, which remarkably strengthened $O_3$ stomatal uptake during daytime and nighttime. Stomatal opening and transpiration are typically assumed to occur specifically during daytime. However, an increasing number of studies have shown the non-negligible effects of unclosed nocturnal stomata and transpiration for a wide range of plant species (Kukal and Irmak, 2022; Schoppach et al., 2020; Tamang et al., 2019; Ramírez et al., 2018; Hoshika et al., 2018). Therefore, how nocturnal plant activities would interact with the significantly increasing nighttime $O_3$ levels in China during recent years (Agathokleous et al., 2023; He et al., 2022) is also worth deeper investigations. Aside from influencing stomata opening, drastic changes in soil humidity also strengthened NO soil emissions, facilitating NO titration of $O_3$ within the canopy and enhancing non-stomatal $O_3$ removals at night. Therefore, drastically

increasing soil moisture simultaneously led to strong increments in $V_d$. Under current climate warming trends, extreme weather events (such as extreme precipitation and drought) have gained in frequency in agricultural areas (Yuan et al., 2016), and its effect on agriculture, $NO_x$ emissions, $O_3$ formation as well as $O_3$ deposition require future attentions.

During the entire wheat growth season, $O_3$ deposition velocity exhibited large fluctuations under changing environmental conditions, with distinct factors determining $V_d$ variability during different wheat growth stages. These key influencing factors and their effects on $O_3$ deposition would also vary with canopy types and ground surface conditions. Aside from environmental conditions, agricultural activities also significantly affect $O_3$ deposition (Mészáros et al., 2009). Therefore, the actual $O_3$ deposition process bears large uncertainties and varies greatly in space and time. More $O_3$ deposition observations over different types of land surfaces and vegetations are urgently needed to facilitate the exploration on $O_3$ dry deposition mechanisms and to optimize current model parameterizations whose results largely deviate from observed $O_3$ dry deposition fluxes in crop growth seasons (Clifton et al., 2020; Hardacre et al., 2015)."

Data availability. Please refer to the ACP data policy to make data available through a publicly available depository or else state reason (to journal editor) for not being able to make data publicly available.

**Response**: Thanks for your suggestion, we uploaded the observation data in the supplementary data.

Supplemental material

Table S1. This table is by no means comprehensive, a good review of different measurements can be found in Clifton et al. (2020), considering referring to this review here, too. This table S1 however is useful as values of vd are listed for direct comparison, consider changing title to "Summary of selected O3 deposition velocity (…)"

**Response**: Thank you, we changed the title to "Summary of selected $O_3$ deposition velocity ($V_d$)."

Figure S4. As night time ozone can be seen to reach zero, the importance of chemical reactions needs to be considered. Adding $NO_x$ (NO and $NO_2$) data here could provide additional information.

**Response**: Thanks for your suggestion, we acknowledge the importance of chemical reactions on $O_3$ concentration and deposition. The changes of NO concentration during the three soil VWC episodes were presented in Figure 10, which corresponded to the changes of $O_3$ concentration in Figure S7, which have been changed to Figure S5. We added the variations of $NO_2$ concentration in Figure S7. During the episodes, nocturnal $O_3$ concentration could drop to near zero during the 10, 12-13, 30 April, and 22 May (Figure S7), which were all accompanied by high NO concentrations (Figure 10), indicating the strong NO titration at night. In addition, higher $NO_2$ levels were detected when nighttime $O_3$ concentrations were at low levels (Figure S7), which would further consumed nighttime $O_3$ through chemical reactions (Li et al., 2024). Thus, the high levels of NO and $NO_2$ indicated more pronounced chemical consumption of $O_3$ at night during the high soil VWC events. The higher $NO_x$ concentration after the watering process might be attributed to enhanced soil NO emission, which would cause stronger NO titration with $O_3$ within wheat canopy and facilitate the non-stomatal $O_3$ deposition at night. We highlighted the importance of nocturnal titration reactions on $O_3$ deposition and concentration in Lines 408-411:

"As shown in Figure 10, high NO became more frequent at night during the high soil VWC events, and nighttime $V_d$ dramatically increased when $NO_x$ (NO and $NO_2$) fluctuated at obviously higher levels and nighttime $O_3$ concentration was still at low level (Figure S7), indicating a more intensive titration consumption of $O_3$ at night."

[Figure]

**Figure S7. Variations of $O_3$ flux (purple lines), concentration (orange lines), NO2 concentration (green lines) during (a) 8-13 April, (b) 27 April-1 May and (c) 18-22 May, with shades representing daytime hours.**

Reference:

Li, J., Wang, S., Yang, T., Zhang, S., Zhu, J., Xue, R., Liu, J., Li, X., Ge, Y., and Zhou, B.: Investigating the causes and reduction approaches of nocturnal ozone increase events over Tai'an in the North China Plain, Atmospheric Research, 307, 107499, https://doi.org/10.1016/j.atmosres.2024.107499, 2024.

**Technical corrections**

Abstract. Revise the first two sentences, they are vague, making not clear how ozone is related to air quality, ecosystems and climate change, and who/what is experiencing increased ozone exposure.

**Response**: Thank you for the comments, we rephrased the two sentences as "Ozone ($O_3$) deposition is the main sink of surface $O_3$, exerting great influences on air quality and ecosystem. Due to instrument limitations and method shortages, $O_3$ deposition was less observed and investigated in China, where $O_3$ has been reported to be on continuous and significant rise."

1 Introduction.

Line 30, delete "etc." as it is too unspecific.

**Response**: Thanks, we deleted it.

Line 34, consider rephrasing second part of the sentence "and in the budget of tropospheric $O_3$" including a verb.

**Response**: We revised this sentence as: "Dry deposition plays one of the key roles in removing surface $O_3$ (e.g., (Tang et al., 2017) and ==contributes about 20% to the annual global tropospheric $O_3$ loss== (Lelieveld and Dentener, 2000; Wild, 2007; Hardacre et al., 2015)"

Line 36, Insert "plant" before "uptake" and substitute "higher" with "large"

**Response**: Thanks, we revised them as suggestions.

Line 37, Substitute "hazardous" with "deleterious"

**Response**: Thanks, we changed it.

Line 40/41, sentence on ozone deposition and its contribution to the tropospheric ozone budget should be merged with similar sentence in line 34/35

**Response**: Thank you for the reminder, we combined the two sentences as "Dry deposition plays one of the key roles in removing surface $O_3$ (e.g., (Tang et al., 2017) and ==contributes about 20% to the annual global tropospheric $O_3$ loss== (Lelieveld and Dentener, 2000; Wild, 2007; Hardacre et al., 2015)"

Line 44, consider rewording, unclear what is exactly meant by "ecological environment"

**Response**: We revised "ecological environment" to "ecosystem".

Line 49, delete "the" before "environmental factors"

**Response**: Thanks, we deleted it.

Line 53, providing a decimal place "31.2%" seems over-precise in this context

**Response**: We kept the same decimal place of the estimated contributions of stomatal $O_3$ deposition in the reference (Xu et al., 2018).

Reference:

Xu, J., Zheng, Y., Mai, B., Zhao, H., Chu, Z., Huang, J., and Yuan, Y.: Simulating and partitioning ozone flux in winter wheat field: the Surfatm-$O_3$ model (in Chinese), China Environmental Science 38, 455-470, 2018.

Line 55, consider adding a reference or two, e.g. Altimir et al., 2006; Clifton et al., 2020

**Response**: Thanks for your suggestion, we added the two references.

Lines 55-57: "Currently, the eddy covariance (EC) method ==and flux-gradient (FG) approach are the most commonly used micrometeorological techniques for measuring O$_3$ vertical fluxes (Businger and Oncley, 1990; Altimir et al., 2006; Wu et al., 2015; Clifton et al., 2020b)=="."

Line 72, change "wildly" to "widely"

**Response**: Thanks, we revised it.

2.2. Relaxed eddy accumulation (REA) technique.

Line 106, add year of measurements, i.e. "February to June 2023"

**Response**: Thanks, we added the year of the measurements.

3.1 Meteorological conditions

Figure 2. the label "u*" on the second panel seems cut off at the bottom

**Response**: Thanks for pointing out it, we modified the label "u*" in Figure 2b.

[Figure]

**Figure 2: Daily meteorological and soil conditions from 12 February to 18 June 2023: (a) T$_{Air}$ and RH, (b) u$_*$ and VPD, (c) ==irrigation (black arrows)==, precipitation, T$_{Soil}$ and soil VWC, (d) G and PAR.**

Supplemental material.

Table S1. Delete "recent years" as measurements/references are from 90s or early 2000s

**Response**: Thanks, we deleted it.

---

## Author Response (AR2)

**Response to the preceding review file validation**

Please ensure that the colour schemes used in your maps and charts allow readers with colour vision deficiencies to correctly interpret your findings. Please check your figures using the Coblis – Color Blindness Simulator (https://www.color-blindness.com/coblis-color-blindness-simulator/) and revise the colour schemes accordingly.=> Figs. 5, 6, 8 I just noticed that your figure S1 contains an aerial. To clarify whether a copyright statement or a credit must be given in the map itself or in the caption, we differentiate between (a) maps entirely created by you, (b) maps created by you but based on layers reused from other originators, or (c) maps simply reused from other originators. An example for (a) is a digital elevation model (DEM) purely based on measurement points collected by you and derived by using a software product. If you use an existing map layer from another originator as a basis for significantly enriching the map with your own content, this would be an example for case (b). Case (c) could be a pure reproduction of Google Maps where your own contribution is rather small (e.g. a city map where you only added a few marks for your study locations). If the map was entirely created by you (case a), there is no need to change the caption or map. Please simply inform us. To the contrary, if your map follows cases (b) or (c), please let us know whether the map is distributed under public domain. If yes, please do not include a copyright statement (copyright is waived) but consider adding a credit to the map or caption. However, if your map follows cases (b) or (c) and is not distributed under public domain, please include at least a credit or even a copyright statement (e.g. © Google Maps), if this is required by the map provider, in the map itself or in the caption.

**Response**: Thanks for your reminders again. We changed the color schemes in Figure 5, 6, 8 according to your suggestions. The map in Figure S1 was generated by ArcGIS and python program written by us, and we added a statement in the figure title of Figure S1.